# Molecular basis for receptor recognition and broad host tropism for merbecovirus MjHKU4r-CoV-1

Zhennan Zhao[1,3], Xin Li[1,3], Yan Chai[1,3], Zhifeng Liu[1,2], Qihui Wang [ID][1] & George F Gao [ID][1,2 ✉]

## Abstract

A novel pangolin-origin MERS-like coronavirus (CoV), MjHKU4r-CoV-1, was recently identified. It is closely related to bat HKU4-CoV, and is infectious in human organs and transgenic mice. MjHKU4r-CoV-1 uses the dipeptidyl peptidase 4 (DPP4 or CD26) receptor for virus entry and has a broad host tropism. However, the molecular mechanism of its receptor binding and determinants of host range are not yet clear. Herein, we determine the structure of the MjHKU4r-CoV-1 spike (S) protein receptor-binding domain (RBD) complexed with human CD26 (hCD26) to reveal the basis for its receptor binding. Measuring binding capacity toward multiple animal receptors for MjHKU4r-CoV-1, mutagenesis analyses, and homology modeling highlight that residue sites 291, 292, 294, 295, 336, and 344 of CD26 are the crucial host range determinants for MjHKU4r-CoV-1. These results broaden our understanding of this potentially high-risk virus and will help us prepare for possible outbreaks in the future.

**Keywords** MjHKU4r-CoV-1; Virus Entry; RBD; CD26; Interspecies Transmission
**Subject Categories** Microbiology, Virology & Host Pathogen Interaction; Structural Biology

## Introduction

Coronaviruses (CoVs) belonging to the *Coronaviridae* family are enveloped, positive-sense, single-stranded RNA viruses that are classified into four genera: Alpha, Beta, Gamma, and Deltacoronavirus (Woo et al, 2012). Three BetaCoVs have been reported to be life-threatening and caused outbreaks, including severe acute respiratory syndrome coronavirus (SARS-CoV) in 2003, the Middle East respiratory syndrome coronavirus (MERS-CoV) in 2012, and severe acute respiratory syndrome coronavirus 2 (SARS-CoV-2) since 2020 (Allen et al, 2017; Jones et al, 2008; Sohrabi et al, 2020) which has posed major threats to public health and caused serious social and economic impacts.

Recurrent coronavirus zoonoses and the detection of numerous coronaviruses in wildlife suggest that cross-species transmission

events are constantly occurring (Tortorici et al, 2022). SARS-CoV and SARS-CoV-2 are generally believed to originate from bats (Hu et al, 2017; Lu et al, 2020; Wang et al, 2021; Tan et al, 2020; Zhu et al, 2020). Bats are also suspected to be the natural reservoir of MERS-CoV (Wang et al, 2014; Memish et al, 2013; Ithete et al, 2013). However, the fact that no virus with a whole genome is highly homologous to MERS-CoV prevents concluding that MERS-CoV originated from bats (Wang et al, 2021; Cui et al, 2019). As the largest reservoir of Alpha and BetaCoVs, bats play pivotal roles in cross-species transmission of coronaviruses (Wang et al, 2014; Li et al, 2005a; Woo et al, 2012). However, rare physical contact between bats and humans occurs, and bat CoVs are normally required to evolve further to acquire the ability to infect humans, suggesting intermediate animal hosts may be needed (Chen et al, 2023; Zhao et al, 2022a). For a while, civets and dromedary camels were regarded as such intermediates for SARS-CoV and MERS-CoV, respectively (The Chinese SARS Molecular Epidemiology Consortium, 2004; Li et al, 2005b; Song et al, 2005; Guan et al, 2003; Haagmans et al, 2014; Reusken et al, 2013). However, this notion might need to be reassessed. Identification of two SARS-CoV-2-related coronaviruses, GD/1/2019 and GX/P2V/2017 (Lam et al, 2020; Xiao et al, 2020), suggests that pangolins have potential as intermediate animal hosts. A recent study isolated a pangolin-origin MERS-like CoV, MjHKU4r-CoV-1, that is related to bat HKU4-CoV but may be capable of directly infecting humans and exhibiting enhanced infectivity to human cells/organs compared to bat viruses (Chen et al, 2023).

Virus entry is initiated by binding of viral particles to host receptors (Dimitrov, 2004). The receptor recognition of coronaviruses is mediated by the spike (S) protein on the viral envelope, which modulates host and tissue tropism, pathogenicity, and zoonotic transmission (Tortorici et al, 2022; Evans and Liu, 2021; Zhao et al, 2023; Zhao et al, 2022b; Belouzard et al, 2012). S protein is composed of S1 and S2 subunits that engage receptors and mediate membrane fusion, respectively (Li, 2016). To date, three host peptidases, including aminopeptidase N (APN), angiotensin-converting enzyme 2 (ACE2), and dipeptidyl peptidase 4 (DPP4 or CD26), have been characterized as the cellular receptors utilized by coronaviruses (Bosch et al, 2014). ACE2 is a major receptor of SARS-CoV and SARS-CoV-2 (Wang et al, 2020; Li et al, 2003). CD26 is "hijacked" by MERS-CoV, HKU4-CoV, and MjHKU4r-CoV-1 for infection (Wang et al, 2014; Chen et al, 2023; Raj et al, 2013).

The binding capacity of these receptors for viral S proteins is one of the crucial factors determining viral infectivity and

[1]CAS Key Laboratory of Pathogen Microbiology and Immunology, Institute of Microbiology, Chinese Academy of Sciences, Beijing 100101, China. [2]College of Veterinary Medicine, Shanxi Agricultural University, Jinzhong 030801, China. [3]These authors contributed equally: Zhennan Zhao, Xin Li, Yan Chai. ✉E-mail: gaof@im.ac.cn

pathogenicity. Previous studies displayed that CD26 interacts with the receptor-binding domain (RBD) of MERS-CoV, not the NTD (Lu et al, 2013; Yuan et al, 2020). Recently reported MjHKU4r-CoV-1 also uses its RBD to interact with CD26 for virus entry (Chen et al, 2023). Thus far, the molecular basis of CD26 recognition by MjHKU4r-CoV-1 remains unknown. Furthermore, previous data demonstrate that MjHKU4r-CoV-1 might have a broad host range (Chen et al, 2023). However, its underlying determinants were not demonstrated, hindering our understanding of the zoonotic risks of this virus.

In this study, we measure the binding affinity between human CD26 (hCD26) and the RBD of MjHKU4r-CoV-1, HKU4-CoV, or MERS-CoV and explore the molecular mechanism of MjHKU4r-CoV-1 RBD binding to hCD26. We then evaluate the receptor binding capability of the MjHKU4r-CoV-1 RBD for 17 other animal CD26 orthologs, covering eight orders, and further identify critical host range determinants by mutagenesis analyses and structural homology modeling. Altogether, our data delineate the molecular basis of receptor recognition and determinants of the broad host tropism of a novel merbecovirus, MjHKU4r-CoV-1, highlighting the importance of in-depth research on these potentially high-risk viruses and providing important molecular evidence for possible outbreaks in the future.

# Results

## The receptor binding affinity and molecular basis of hCD26 recognition by MjHKU4r-CoV-1

A recent study shows that MjHKU4r-CoV-1 has the potential to infect humans (Chen et al, 2023). MjHKU4r-CoV-1 shares 84.63% and 66.23% amino acid (AA) sequence identity with HKU4-CoV and MERS-CoV in their S proteins, respectively. The AA sequence of the MjHKU4r-CoV-1 RBD has 78.75% and 58.92% identity to that of the HKU4-CoV and MERS-CoV RBDs, respectively (Fig. 1A). Receptor binding is a key aspect of achieving viral infectivity. To evaluate the receptor binding capacity of MjHKU4r-CoV-1, the binding affinities between hCD26 and RBDs from MjHKU4r-CoV-1, HKU4-CoV, and MERS-CoV were analyzed by surface plasmon resonance (SPR). The affinity of the MjHKU4r-CoV-1 RBD to hCD26 was $0.41 \pm 0.03\,\mu M$ (Fig. 1B). The $K_D$ of HKU4-CoV RBD binding to hCD26 was approximately $5.62 \pm 2.66\,\mu M$, which is ~14-fold lower than that of the MjHKU4r-CoV-1, while MERS-CoV RBD $(1.21 \pm 0.03\,\mu M)$ has a similar binding capacity for hCD26 to that of MjHKU4r-CoV-1 RBD (Fig. 1B).

To unravel the underlying molecular mechanism of the S protein of MjHKU4r-CoV-1 binding to hCD26, we determined the atomic structures of the MjHKU4r-CoV-1 RBD in complex with hCD26 by X-ray crystallography at a resolution of 3.5 Å (Fig. 1C and EV1, and Table 1). Structural analysis revealed that residues Y463, K509, H517, N518, E521, S543, D545, E547, and R550 of the MjHKU4r-CoV-1 RBD form an extensive H-bond network with residues K267, T288, A289, A291, L294, R317, R336, and Q344 of hCD26 (Fig. 1D, Table 2). Notably, the N-glycans of N229 on hCD26 were observed to form more extensive hydrophilic interactions with the MjHKU4r-CoV-1 RBD (Fig. 1D) than that observed in the HKU4-CoV RBD/hCD26 and MERS-CoV RBD/

hCD26 complex structures (Wang et al, 2014; Lu et al, 2013). Furthermore, there is a hydrophobic pocket formed by residues V513, V548, V565, I561, and V563 of the MjHKU4r-CoV-1 RBD, and a helix on hCD26 formed by A289, P290, A291, S292, M293, L294, I295, and G296 inserts into this groove. Residues A291, L294, and I295 of hCD26 form extensive hydrophobic interactions with the hydrophobic pocket (Fig. 1E).

## Mutagenesis analyses of the key residues for hCD26 binding to the MjHKU4r-CoV-1 RBD

Structural analysis revealed that the side chains of residues K267, T288, R317, R336, and Q344 and the N-glycans of N229 on hCD26 are hydrogen-bonded to the MjHKU4r-CoV-1 RBD (Fig. 1D). Therefore, mutagenesis analyses for the above six residues (K267A, T288A, R317A, R336A, Q344A, and N229A/Q) was further performed to determine whether these residues are dominant for receptor binding. However, the two N229 mutants (N229A and N229Q) of hCD26 could not be produced. SPR results suggest that the K267A substitution in hCD26 caused an ~10.4-fold decrease in binding affinity toward the MjHKU4r-CoV-1 RBD (Fig. 2A,B). The binding affinity for the MjHKU4r-CoV-1 RBD toward hCD26 decreased ~21.1-fold when R317 of hCD26 was mutated to A317 (Fig. 2A,C). The R336A substitution in hCD26 also reduced the binding affinity for the MjHKU4r-CoV-1 RBD (Fig. 2A,D). The binding affinity between the MjHKU4r-CoV-1 RBD and hCD26 harboring T288A or Q344A mutations was ~0.35 μM or 0.28 μM, respectively (Fig. 2E,F), similar to that between the wild-type MjHKU4r-CoV-1 RBD and hCD26 (Fig. 2A). The above results indicate that the side chains of residues K267, R317 and R336 are crucial for hCD26 binding to the MjHKU4r-CoV-1 RBD.

## Structural comparisons of MjHKU4r-CoV-1, HKU4-CoV, and MERS-CoV RBDs in complex with hCD26

Previous work demonstrates the molecular mechanism of viral entry of HKU4- and MERS-CoVs using hCD26 (Wang et al, 2014; Lu et al, 2013). To reveal why MjHKU4r-CoV-1 has stronger binding affinities than HKU4-CoV and MERS-CoV, we analyzed the features in the interface between the RBDs and hCD26. Due to the AA distinctions among the MjHKU4r-CoV-1, HKU4-CoV, and MERS-CoV RBDs, the RBD/hCD26 interfaces display significant differences (Fig. 3A–C). For convenient structural comparison, the AA sequences of the HKU4-CoV and MERS-CoV RBDs were renumbered based on that of the MjHKU4r-CoV-1 RBD, as shown in the AA sequence alignment among the MjHKU4r-CoV-1, HKU4-CoV, and MERS-CoV RBDs (Fig. 1A). Structural comparisons revealed that although there are many shared key residue sites among these two or three RBDs, several shared sites are occupied by different AA residues (Fig. 3A–D), which may affect their interactions toward CD26. Among these, residues H517 and R550 of the MjHKU4r-CoV-1 RBD have the most evident contact increase than N517 and K550 of the HKU4-CoV RBD, showing 21 and 13 contact differences, respectively (Fig. 3A,B). Compared with the MERS-CoV RBD, residues H517 and S519 of MjHKU4r-CoV-1 RBD could contribute to higher binding capacity. In addition, there are many unique sites for each RBD to engage with hCD26 (Fig. 3D). Notably, the interface formed by residues S502, F537, S538, Y540, G541, F542 and F549 of the MjHKU4r-CoV-1 RBD

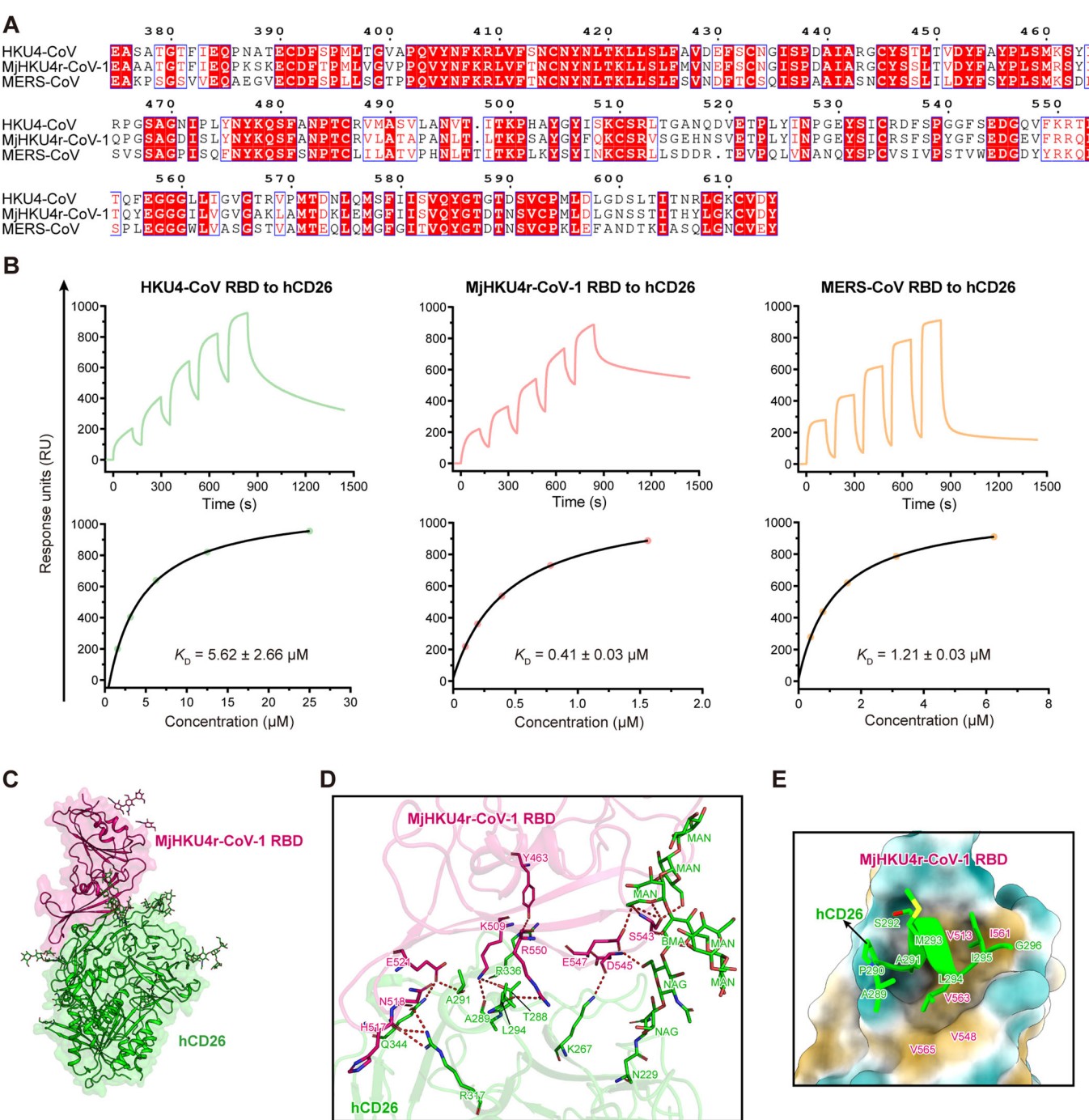

Figure 1. Receptor-binding characteristics of MjHKU4r-CoV-1.

(A) AA sequence alignment of the RBDs from HKU4-CoV, MjHKU4r-CoV-1, and MERS-CoV. (B) The SPR curves for HKU4-CoV, MjHKU4r-CoV-1, and MERS-CoV bound to hCD26 are shown. Raw curves are shown with different colors, as indicated. The fit curves are represented by black lines. $K_D$ values are the mean ± SD of three biological replicates. (C) The overall architecture of the MjHKU4r-CoV-1 RBD/hCD26 crystal structure. (D) Interaction network in the MjHKU4r-CoV-1 RBD/hCD26 complex. Side chains of interacting residues on hCD26 (green) and the MjHKU4r-CoV-1 RBD (hot pink) are shown as sticks and labeled appropriately. The red dashes present H-bonds. (E) Hydrophobic interactions between the MjHKU4r-CoV-1 RBD and hCD26. The MjHKU4r-CoV-1 RBD is shown as a surface according to hydrophobicity (green: hydrophilic; white: neutral; and gold: hydrophobic). The helix (residues 289–296) from hCD26 is shown as a cartoon and sticks and colored in green. Source data are available online for this figure.

**Table 1.  X-ray data collection and refinement statistics.**

|  | MjHKU4r-CoV-1 RBD/hCD26 |
|---|---|
| *Data collection* |  |
| Space group | P3₂21 |
| Cell dimensions |  |
| *a, b, c* (Å) | 127.501, 127.501, 245.633 |
| α, β, γ (°) | 90.00, 90.00, 120.00 |
| Resolution(Å) | 50.00-3.50 (3.71-3.5) |
| Unique reflections | 54,034 (6746) |
| Completeness (%) | 95.9 (74.7) |
| $R_{meas}$ | 0.231 (1.566) |
| *I*/σ*I* | 10.31 (1.77) |
| CC$_{1/2}$ (%) | 0.995 (0.625) |
| Redundancy | 7.0 (6.7) |
| *Refinement* |  |
| Resolution (Å) | 45.78-3.50 |
| No. reflections | 28,683 |
| $R_{work}$/$R_{free}$ | 0.2107/0.2375 |
| No. atoms |  |
| Protein | 8046 |
| Ligand/ion | 0 |
| Water | 0 |
| *B*-factors (Å²) |  |
| Protein | 104.80 |
| Ligand/ion | 148.49 |
| Water |  |
| R.M.S. deviations |  |
| Bond lengths (Å) | 0.004 |
| Bond angles (°) | 0.679 |
| Ramchandran Statistics (%) |  |
| Favored | 96.67 |
| Allowed | 3.33 |
| Disallowed | 0 |

Values in parentheses are for the highest-resolution shell.

interacts with the N-glycans of N229 in hCD26, which is not observed in the HKU4-CoV RBD/hCD26 and MERS-CoV RBD/hCD26 complexes due to the shorter N-glycans resulting from different protein expression systems. In this work, a mammalian expression system was used for hCD26 expression. However, the hCD26 protein for the HKU4-CoV RBD/hCD26 and MERS-CoV RBD/hCD26 complexes was expressed using the bac-to-bac baculovirus system, which produces simpler sugar modifications. Residue Y540 of the MjHKU4r-CoV-1 RBD has a longer side chain than G540 of the HKU4-CoV RBD and S540 of the MERS-CoV RBD (Fig. 1A), potentially contributing to a higher binding affinity toward N-glycans of N229 on hCD26.

The interaction area (excluding N-glycans) of hCD26 in the MjHKU4r-CoV-1 RBD/hCD26 complex is smaller than that of the

HKU4-CoV RBD/hCD26 and MERS-CoV RBD/hCD26 complexes (Fig. 3E), and contains the fewest van der Waals contacts (VDWs). However, the structural alignment shows that the MjHKU4r-CoV-1 RBD is closer to hCD26 and its N229 N-glycans than the HKU4-CoV and MERS-CoV RBDs (Fig. EV2), suggesting that the MjHKU4r-CoV-1 RBD forms stronger interactions to bind more tightly to hCD26 than the other two RBDs.

The polar interaction network analysis was further performed. For convenient comparison, interaction networks were divided into four patches: patch1 (Fig. 4A), patch2 (Fig. 4B), patch3 (Fig. 4C), and patch4 (Fig. 4D). The loop (residues 514–520) is less conserved among the HKU4-CoV, MjHKU4r-CoV-1, and MERS-CoV RBDs (Fig. 1A), which causes conformational distinctions among these three merbecoviruses to form different interaction networks with CD26 (Figs. 4A and EV2). In the HKU4-CoV RBD/hCD26 complex, residue R317 of hCD26 engages N517 and Q518 of the RBD with three H-bonds (Fig. 4A). In addition, D519 of the HKU4-CoV RBD interacts with Y322 of hCD26 (Fig. 4A). In the MjHKU4r-CoV-1 RBD/hCD26 complex, residue R317 of hCD26 forms three H-bonds with H517 and N518 of the RBD (Fig. 4A). However, R317 of hCD26 is only hydrogen-bonded with D517 of the MERS-CoV RBD, which also forms an H-bond with Y322 of hCD26 (Fig. 4A). Residue E521 of the MjHKU4r-CoV-1 and MERS-CoV RBDs forms H-bonds with Q344 and A291 of hCD26 (Fig. 4A), which was not observed in the HKU4-CoV RBD/hCD26 complex (Fig. 4A).

K509 in the RBD is conserved in the three complexes and forms two H-bonds with T288 and A289 of hCD26 (Fig. 4B). R550 in the MjHKU4r-CoV-1 and MERS-CoV RBDs interacts with L294 of hCD26 (Fig. 4B), while residue 550 is K in the HKU4-CoV RBD and forms an H-bond with I295, not L294 (Fig. 4B). Notably, residue N508 of the MERS-CoV RBD is hydrogen-bonded with Q286 of hCD26, which is not observed in the other two complexes (Fig. 4B).

D545 in the MjHKU4r-CoV-1, HKU4-CoV, and MERS-CoV RBDs forms an H-bond with K267 of hCD26 (Fig. 4C). K267 of hCD26 also forms an extra H-bond with E544 of the HKU4-CoV RBD or D547 of MERS-CoV RBD (Fig. 4C). Residue 547 of the HKU4-CoV RBD (Q547) or MjHKU4r-CoV-1 (E547) differs from that of the MERS-CoV RBD (D547) (Fig. 4C). Q547 of the HKU4-CoV RBD engages with N-glycans of N229 together with E544 of the RBD, while E547 of the MjHKU4r-CoV-1 RBD interacts with N-glycans of N229 together with S543 of the RBD (Fig. 4C). Notably, no residues of the MERS-CoV RBD were observed to form H-bonds with the N-glycans of N229 on CD26. In contrast, there is an additional H-bond between T265 of hCD26 and Y548 of the MERS-CoV RBD (Fig. 4C).

In the HKU4-CoV RBD/hCD26 complex, R336 forms an H-bond with N471 of the RBD (Fig. 4D). However, residue Y463, not D471, in the MjHKU4r-CoV-1 RBD is hydrogen-bonded to R336 of hCD26 (Fig. 4D). Residue 463 on the MERS-CoV RBD is D, which is hydrogen-bonded to R336 together with residue Y506 (Fig. 4D). In addition, the MjHKU4r-CoV-1 RBD has a hydrophobic pocket similar to HKU4-CoV and MERS-CoV (Fig. 4E). Altogether, due to AA distinctions among MjHKU4r-CoV-1, HKU4-CoV, and MERS-CoV RBDs, MjHKU4r-CoV-1 RBD forms a different interaction network, allowing it to more tightly bind to hCD26 than the other two RBDs.

**Table 2.** Amino acid residues comparison of the MjHKU4r-CoV-1, HKU4-CoV, and MERS-CoV RBDs interacting with hCD26.

| hCD26 | MjHKU4r-CoV-1 RBD | HKU4-CoV RBD | MERS-CoV RBD |
|---|---|---|---|
| T188 | – | – | E544 (1) |
| T265 | – | – | Y548 (3, **1**) |
| V266 | – | – | Y548 (1) |
| K267 | D545 (3, **1**), G546 (6) | E544 (5, **1**), D545 (5, **1**), G546 (6) | E544 (3), D545 (5, **1**), G546 (12), D547 (7, **1**) |
| F269 | – | D545 (2) | D545 (5) |
| Q286 | – | D545 (1), G546 (4), T567 (1) | N508 (4, **1**), D545 (1), G546 (11), S565 (3), G566 (1), S567 (1) |
| T288 | Q508 (2), K509 (5, **1**) | S508 (1), K509 (5, **1**), V565 (2) | N508 (4), K509 (10, **1**), S565 (4) |
| A289 | K509 (3, **1**) | K509 (4, **1**) | K509 (5, **1**) |
| P290 | K509 (3), E521 (3) | K509 (3), E521 (2) | K509 (3), E521 (3) |
| A291 | K509 (1), S511 (1), V513 (5), E521 (10, **1**), V563 (4) | K509 (3), L513 (7), E521 (9), I563 (5) | K509 (2), S511 (1), L513 (4), E521 (11, **1**), V563 (4) |
| S292 | V513 (2), N518 (4), S519 (1), E521 (1) | L513 (2), Q518 (2), D519 (2) | L513 (7), D517 (1) |
| L294 | K509 (2), V548 (4), R550 (6, **1**), V563 (3), V565 (1) | K509 (1), V548 (4), I563 (2), V565 (1) | K509 (2), Y548 (7), R550 (8, **1**), V563 (4) |
| I295 | V513 (3), N518 (4), R550 (10), I561 (2), V563 (1) | L513 (1), Q518 (6), K550 (9, **1**), L561 (2), I563 (4) | L513 (3), D517 (3), R550 (12), W561 (28), V563 (2) |
| G296 | R550 (6) | – | R550 (7) |
| D297 | – | – | R550 (1) |
| H298 | – | – | Y548 (7) |
| R317 | H517 (6, **2**), N518 (7, **1**) | N517 (5, **2**), Q518 (5, **1**) | L513 (1), D517 (11, **2**) |
| Y322 | H517 (4), N518 (1), S519 (8) | N517 (1), D519 (7, **1**) | D517 (4, **1**), R518 (4) |
| E332 | – | Y463 (1) | – |
| S333 | – | Y463 (2), S468 (4) | – |
| S334 | Y463 (1) | Y463 (7), A469 (5), G470 (6) | S462 (1), D463 (1) |
| G335 | – | Y463 (8) | – |
| R336 | Y463 (11, **1**), D471 (4), Y506 (3) | G470 (1), N471 (11, **1**), L474 (2), Y475 (1) | M460 (1), D463 (8, **1**), P471 (9), Y506 (10, **1**) |
| V341 | E521 (7), P523 (2) | E521 (4), P523 (2) | E521 (9), P523 (2) |
| Q344 | E521 (4, **1**) | E521 (1) | E521 (5, **1**) |
| I346 | S519 (2) | D519 (4) | R518 (11) |
| M348 | H517 (3) | N517 (2) | – |
| K392 | – | – | R518 (2) |
| NAG831 | E544 (1) | E544 (14, **1**) | W543 (3), E544 (13), D547 (1) |
| NAG832 | S543 (3), E544 (6), E547 (8, **1**) | S543 (2), E544 (8), Q547 (7, **1**) | W543 (23), D547 (2) |
| BMA833 | S543 (6) | S543 (1), Q547 (1) | W543 (6) |
| MAN836 | F537 (1), S538 (7), G541 (1), F542 (2), S543 (10, **3**), E547 (5, **1**), F549 (2) | – | – |
| MAN837 | S502 (3), Y540 (9), G541 (7), F542 (4), S543 (9, **1**) | – | – |
| MAN838 | Y540 (7) | – | – |
| NAG851 | H517 (17) | N517 (1) | – |
| Total | 267, **16** | 214, **12** | 328, **14** |

The AA sequences of the HKU4-CoV and MERS-CoV RBDs were renumbered based on the MjHKU4r-CoV-1 RBD for convenient comparison. The numbers in parentheses for the MjHKU4r-CoV-1, HKU4-CoV, and MERS-CoV RBDs' residues represent the number of VDWs between the indicated residues with hCD26. The underlined numbers suggest the number of potential H-bonds between the pairs of residues. VDWs were analyzed at a cutoff of 4.5 Å, and H-bonds were calculated at a cutoff of 3.5 Å.

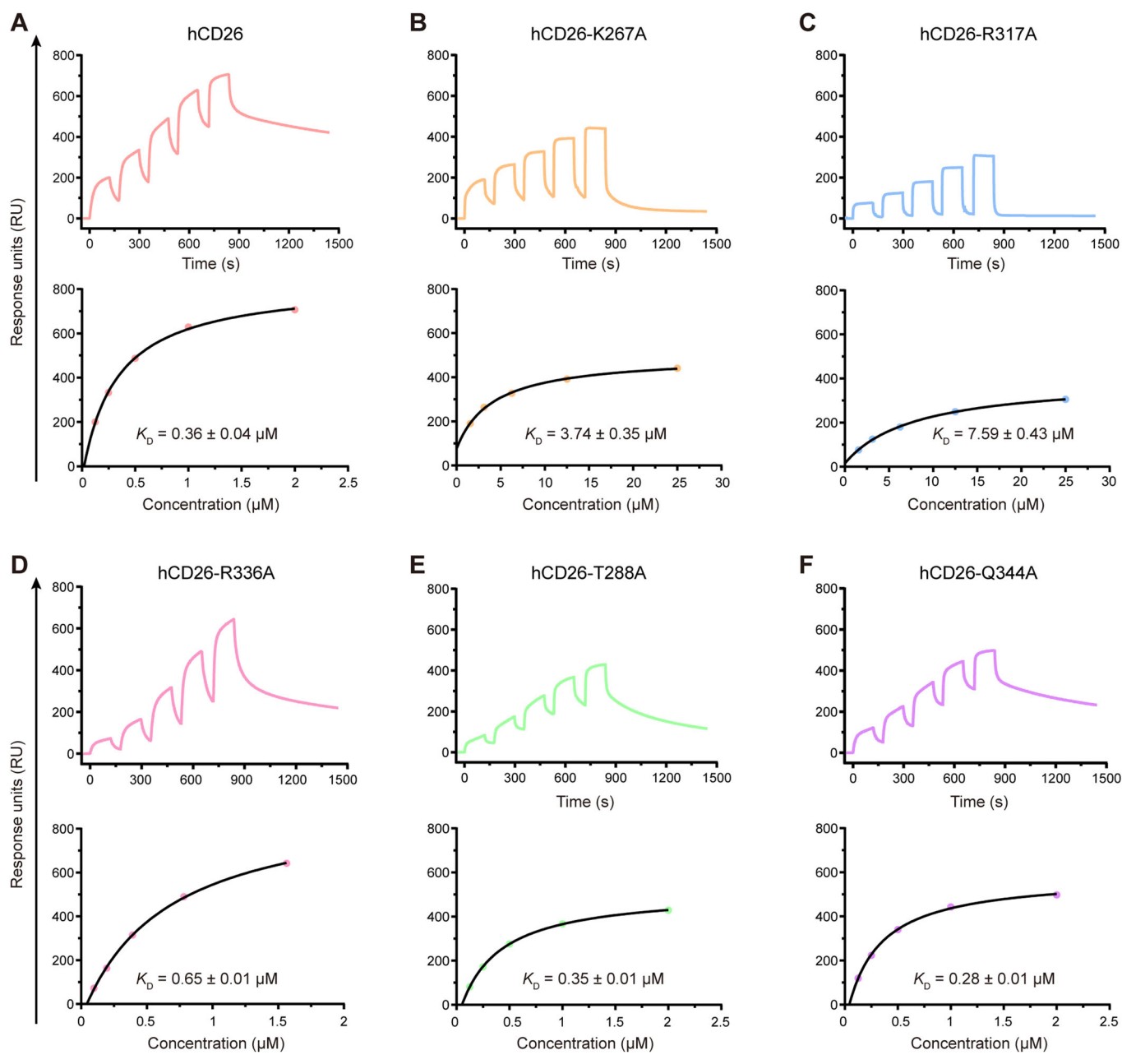

**Figure 2. Mutational analysis of key residues responsible for MjHKU4r-CoV-1 RBD binding.**

(A–F) SPR analysis of binding between the MjHKU4r-CoV-1 RBD and hCD26 (**A**) and its four mutants: hCD26-K267A (**B**), R317A (**C**), R336A (**D**), T288A (**E**), and Q344A (**F**). Raw curves are shown with different colors, as indicated. The fit curves are represented by black lines. $K_D$ values are the mean ± SD of three biological replicates. Source data are available online for this figure.

## The receptor binding spectra of MjHKU4r-CoV-1

Previous work suggests that MjHKU4r-CoV-1 has broad-species receptor binding (Chen et al, 2023). To explore the binding capacity of MjHKU4r-CoV-1 for various animal species, we performed SPR assays to evaluate its receptor binding capacities for 18 animal CD26 orthologs covering Primates (human, macaque, and marmoset), Lagomorpha (rabbit), Pholidota (Malayan pangolin), Perissodactyla (horse), Artiodactyla (goat, sheep, pig, cattle, and camel), Chiroptera (*Pipistrellus pipistrellus* (*Pp*) bat and *Tylonycteris pachypus* (*Tp*) bat), Rodentia (rat and hamster), and Carnivora (cat, dog, and ferret) (Fig. 5). We found that the MjHKU4r-CoV-1 RBD has similar binding affinities to CD26s from human, macaque, marmoset, rabbit, Malayan pangolin, horse, goat, sheep, pig, cattle, and camel (Fig. 5). It displayed a ~6.1-fold affinity decrease for *Pp* bat CD26 compared to hCD26 (Fig. 5). However, it showed no binding capacity for the CD26s from *Tp* bat, rat, hamster, cat, dog, and ferret (Fig. 5).

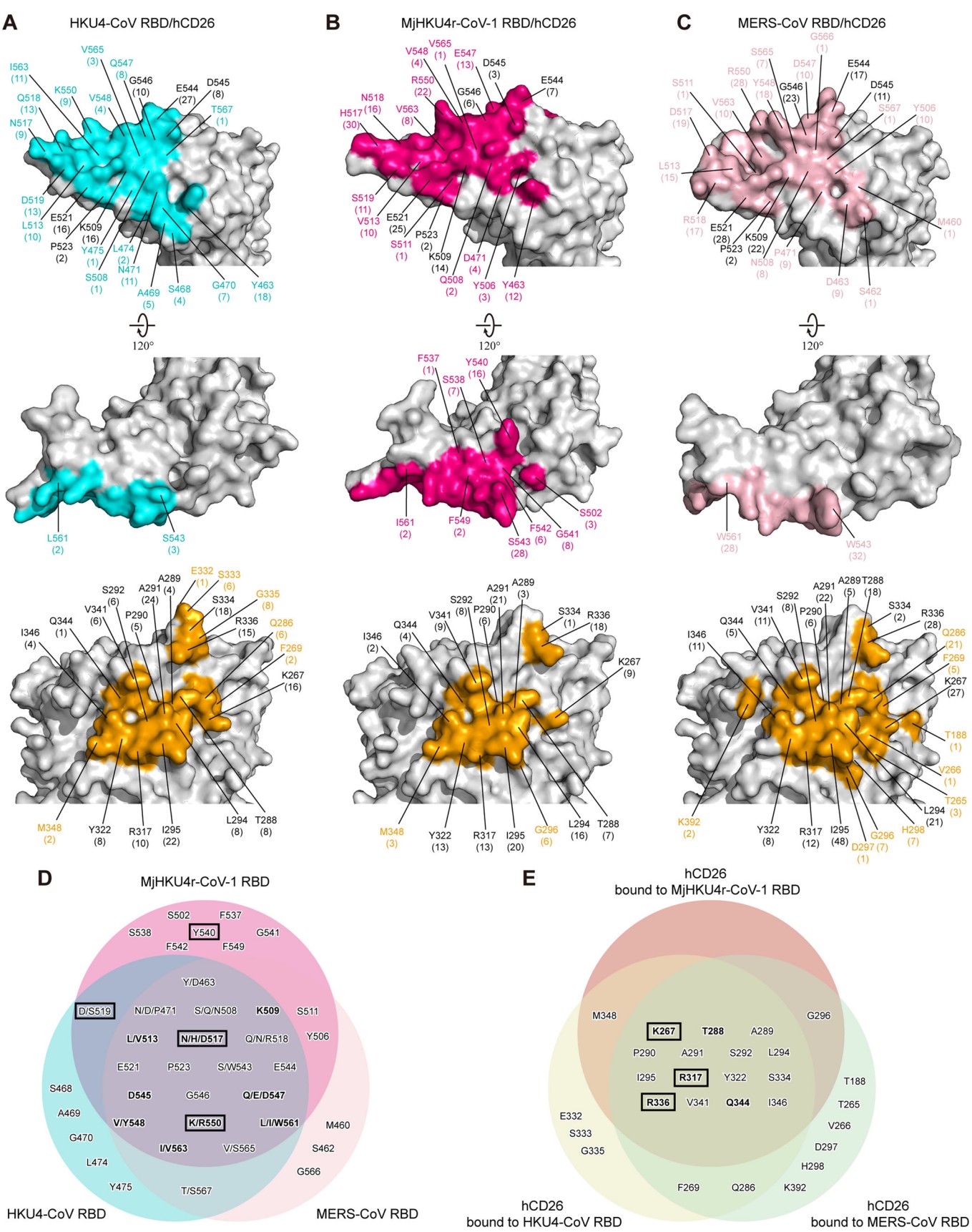

**Figure 3.  Interface comparison among the RBDs of HKU4-CoV, MjHKU4r-CoV-1, and MERS-CoV with hCD26.**

(A–C) The binding interface of HKU4-CoV RBD/hCD26 (PDB: 4QZV) (Wang et al, 2014) (A) MjHKU4r-CoV-1 RBD/hCD26 (B), and MERS-CoV RBD/hCD26 (PDB: 4KR0) (Lu et al, 2013) (C). Three RBDs and hCD26 are shown as surfaces, and interaction areas are colored in cyan (HKU4-CoV RBD), hot pink (MjHKU4r-CoV-1 RBD), pink (MERS-CoV RBD) and orange (hCD26), respectively. The AA sequences of HKU4-CoV and MERS-CoV RBDs are renumbered based on the MjHKU4r-CoV-1 RBD for convenient structural comparison. Interacting residues on the HKU4-CoV, MjHKU4r-CoV-1, and MERS-CoV RBDs are labeled, among which only the same AA residues shared by these three RBDs are colored in black; otherwise in cyan, hot pink, and pink, respectively. Similarly, interacting residues on hCD26 shared by three complexes are labeled in black, otherwise in orange. The numbers of the VDWs for each residue are in parentheses. (D) Venn diagrams of interacting residues on the HKU4-CoV, MjHKU4r-CoV-1, and MERS-CoV RBDs for hCD26. Residues involved in forming H-bonds or hydrophobic interactions with hCD26 and shared by three RBDs are marked in bold. Residues that potentially confer MjHKU4r-CoV-1 RBD with a higher binding affinity are highlighted with black boxes. (E) Venn diagrams of residues on hCD26 bound to the HKU4-CoV, MjHKU4r-CoV-1, and MERS-CoV RBDs. Residues tested in Fig. 2 are marked in bold, and key residues among them are further highlighted with black boxes.

## Host range determinants for MjHKU4r-CoV-1

To reveal the determinants of the host range of MjHKU4r-CoV-1, we aligned the AA sequences of CD26 orthologs from the 18 species tested above, and the hCD26 sites that interact with the MjHKU4r-CoV-1 RBD were labeled (Fig. 6A). For convenient comparison, the AA sequences of the other 17 animal CD26s were renumbered based on that of the hCD26 sequence shown in the alignment (Fig. 6A). Based on the distinct interacting residues between hCD26 and six animal CD26s from *Tp* bat, rat, hamster, cat, dog, and ferret, which display no binding to MjHKU4r-CoV-1, 29 single-site mutations (T288S, A289P, A291E/P/D, S292A/D, L294T/S, I295K/T, S334D/N/T, R336V/T/K/G/S, V341T/L/A/S/E, Q344E/K, I346V/T, and M348T) for hCD26 were generated, and SPR analyses were performed (Figs. 6B and EV3, Table 3). Binding affinity measurements showed that hCD26-I295K, hCD26-L294T, hCD26-A291E, hCD26-S292D, and hCD26-A291D, which individually exist in CD26s from *Tp* bat, rat, hamster, dog, and ferret, do not bind to the MjHKU4r-CoV-1 RBD (Fig. 6B, Table 3). However, all single substitutions in the cat CD26 do not display the loss of binding to the MjHKU4r-CoV-1 RBD (Fig. EV3, Table 3). To determine why cat CD26 did not bind to the MjHKU4r-CoV-1 RBD, a series of multi-site mutants were further constructed (Table 3). The hCD26-I295T-R336K double substitution significantly decreased the binding capacity for the MjHKU4r-CoV-1 RBD, as observed in other double substitutions (hCD26-I295T-R336V, hCD26-I295T-R336T, hCD26-I295T-R336G, and hCD26-I295T-R336S) (Fig. EV3, Table 3). The hCD26-S292A-I295T-R336K three-site substitution has a further decreased binding affinity compared with the hCD26-I295T-R336K mutant, and the hCD26-S292A-I295T-R336K-Q344E mutant almost loses its binding ability (>50 μM) (Figs. 6B and EV3, Table 3). Thus, the lack of binding affinity of cat CD26 for the MjHKU4r-CoV-1 RBD is the comprehensive impact caused by differential AAs, mainly contributed by residues A292, T295, K336, and E344 at the RBD-CD26 interface. The above results implicate that residue sites 291, 292, 294, 295, 336, and 344 of CD26 are crucial for determining their capacity to bind the MjHKU4r-CoV-1 RBD. In those animal CD26s (macaque, marmoset, rabbit, Malayan ganglion, horse, goat, sheep, pig, cattle, and camel) that are comparable to hCD26 for binding the MjHKU4r-CoV-1 RBD, residue sites 291 (A/G), 292 (S), 294 (L), 295 (I), 336 (R), and 344 (Q) are conserved (Fig. 6A).

To decipher how these determinant AAs impact the binding capacity of CD26s from *Tp* bat, rat, hamster, cat, dog, and ferret for the MjHKU4r-CoV-1 RBD, homology modeling was performed (Fig. 7A,B). All models from the above six species displayed high

similarity to hCD26, with a root mean square deviation (RMSD) range of 0.108 to 0.131. Among the six structures, high coverage ranging from 99–100% and high model quality estimation, including Global Model Quality Estimation (GMQE), global absolute quality estimates on the basis of one single model (QMEANDisCo Global), and Quaternary Structure Quality Estimation (QSQE) (Table 4), indicated a lack of major conformational changes between species and supported the validity of using hCD26 as a template for modeling these CD26 interface residues across species. AA sequence alignment showed that residue 295 in the CD26 from human, macaque, marmoset, rabbit, Malayan pangolin, horse, goat, sheep, pig, cattle, and camel is the hydrophobic residue isoleucine (I). However, residue 295 is the hydrophilic residue K in the *Tp* bat CD26 and threonine (T) in the CD26 from *Pp* bat, rat, hamster, cat, dog, and ferret (Fig. 6A). In our MjHKU4r-CoV-1 RBD/hCD26 structure, I295 forms extensive hydrophobic interactions with V513, I561, and V563 of the MjHKU4r-CoV-1 RBD (Fig. 1E). I295K or I295T substitutions would disrupt these hydrophobic interactions, and residue K has a longer side chain, forming steric clashes with the MjHKU4r-CoV-1 RBD, both of which lead to the reduction even ablation of the binding capacity for Mj-RBD-CoV-1, consistent with the SPR results (Figs. 6B, 7A and EV3, Table 3). Similarly, T294 and S294 observed in the rat and ferret CD26s would also impair the hydrophobic interactions with residues V548, V563, and V565 of the MjHKU4r-CoV-1 RBD, and T294 may have steric clashes with the MjHKU4r-CoV-1 RBD, thereby abolishing binding (Figs. 6B, 7A and EV3, Table 3). In addition, residue 291 in the CD26 from hamster, dog, and ferret is E291, P291, and D291, respectively. Both E291 and D291 have a longer side chain that forms steric hindrance with the MjHKU4r-CoV-1 RBD to abrogate binding (Figs. 6B and 7A, Table 3). Although residue P291 has little impact on the dog CD26 for binding to the MjHKU4r-CoV-1 RBD, its neighboring residue D292 forms severe steric clashes to block binding to the MjHKU4r-CoV-1 RBD (Figs. 6B and 7A, Table 3). By contrast, A292 has a somewhat shorter side chain than S292, which slightly decreases the binding capacity (Fig. 7A and EV3, Table 3).

The side chain of R336 in hCD26 forms an H-bond with Y463 and many VDWs with Y463, D471, and Y506 of the MjHKU4r-CoV-1 RBD (Fig. 1D, Table 2). V336 in the rat CD26, T336 in the hamster CD26, K336 in the cat CD26, G336 in the dog CD26, and S336 in the ferret CD26 all have a shorter side chain, which could decrease the binding capacity for the MjHKU4r RBD (Fig. 7B), consistent with the SPR results (Fig. EV3, Table 3). Notably, Q344 of hCD26 is hydrogen-bonded with E521 in the MjHKU4r-CoV-1 RBD/hCD26 complex (Fig. 7B). E344 in the CD26s from rat, cat,

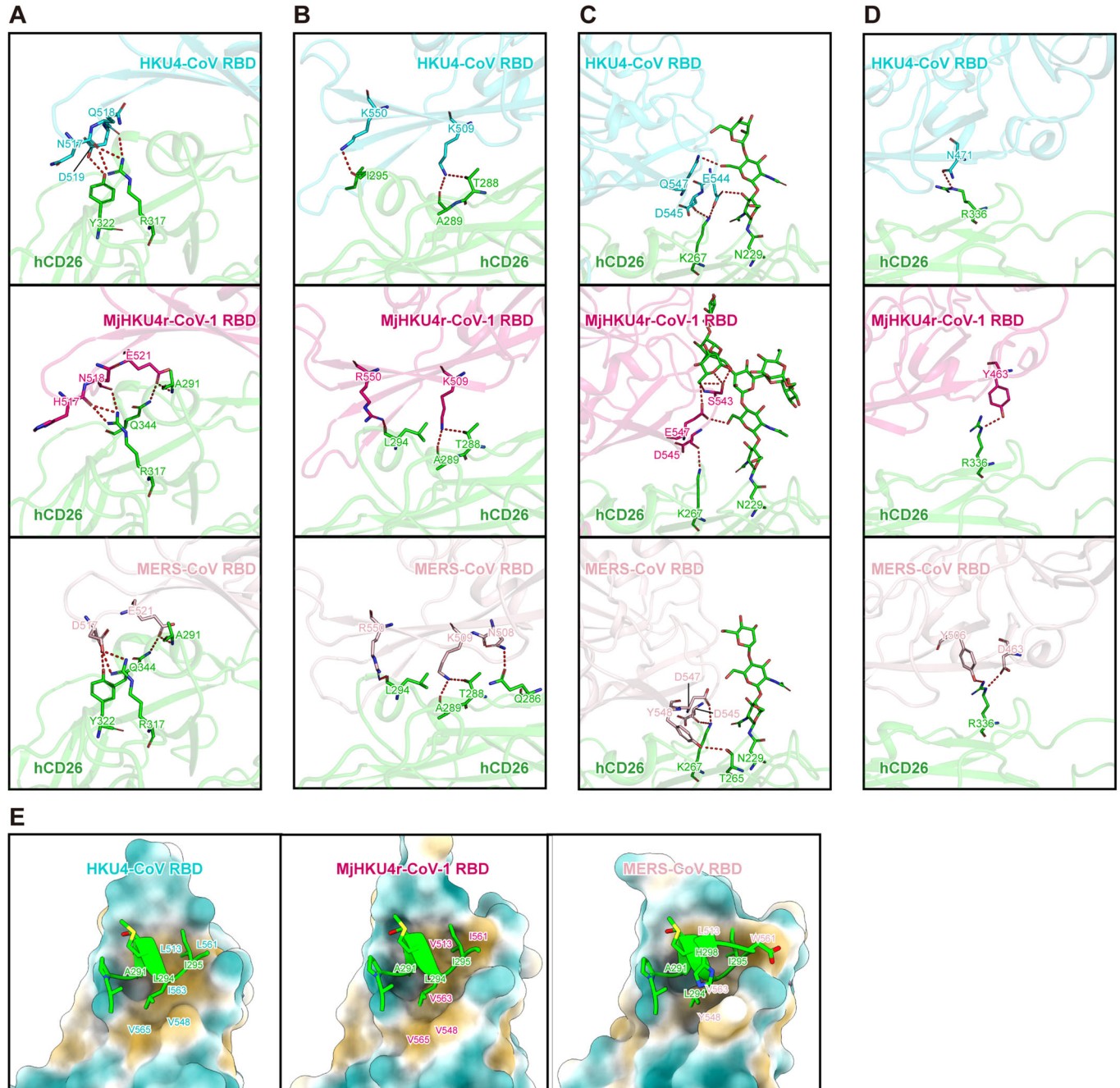

**Figure 4. Structural details of the HKU4-CoV RBD/hCD26, MjHKU4r-CoV-1 RBD/hCD26, and MERS-CoV RBD/hCD26 complexes.**

(A–D) These three complexes were structurally aligned, and their interaction networks were divided into four patches: patch1 (**A**), patch2 (**B**), patch3 (**C**), and patch4 (**D**) for comparison analyses. The AA sequences of the HKU4-CoV and MERS-CoV RBDs were renumbered for convenient structural comparison based on the MjHKU4r-CoV-1 RBD. hCD26 is colored in green, and the HKU4-CoV, MjHKU4r-CoV-1, and MERS-CoV RBDs are colored in cyan, hot pink, and pink, respectively. The red dashes present H-bonds. (**E**) Hydrophobic interactions between the HKU4-CoV, MjHKU4r-CoV-1, or MERS-CoV RBD and hCD26. The HKU4-CoV, MjHKU4r-CoV-1, and MERS-CoV RBDs are shown as surfaces according to hydrophobicity (green: hydrophilic; white: neutral; and gold: hydrophobic). The helix from hCD26 is shown as a cartoon and sticks and colored in green.

and ferret may change the surface electrostatic properties of CD26 to repulse E521 of MjHKU4r-CoV-1 RBD to push themselves away from the interface, thus decreasing the binding affinity between CD26 and MjHKU4r-CoV-1 RBD, as observed in the SPR results (Fig. 6B and EV3, Table 3).

## Discussion

Carbohydrates play pivotal roles in molecule interactions. Like other merbecoviruses (Wang et al, 2014; Lu et al, 2013), the MjHKU4r-CoV-1 S protein interacts with N-glycans on the hCD26

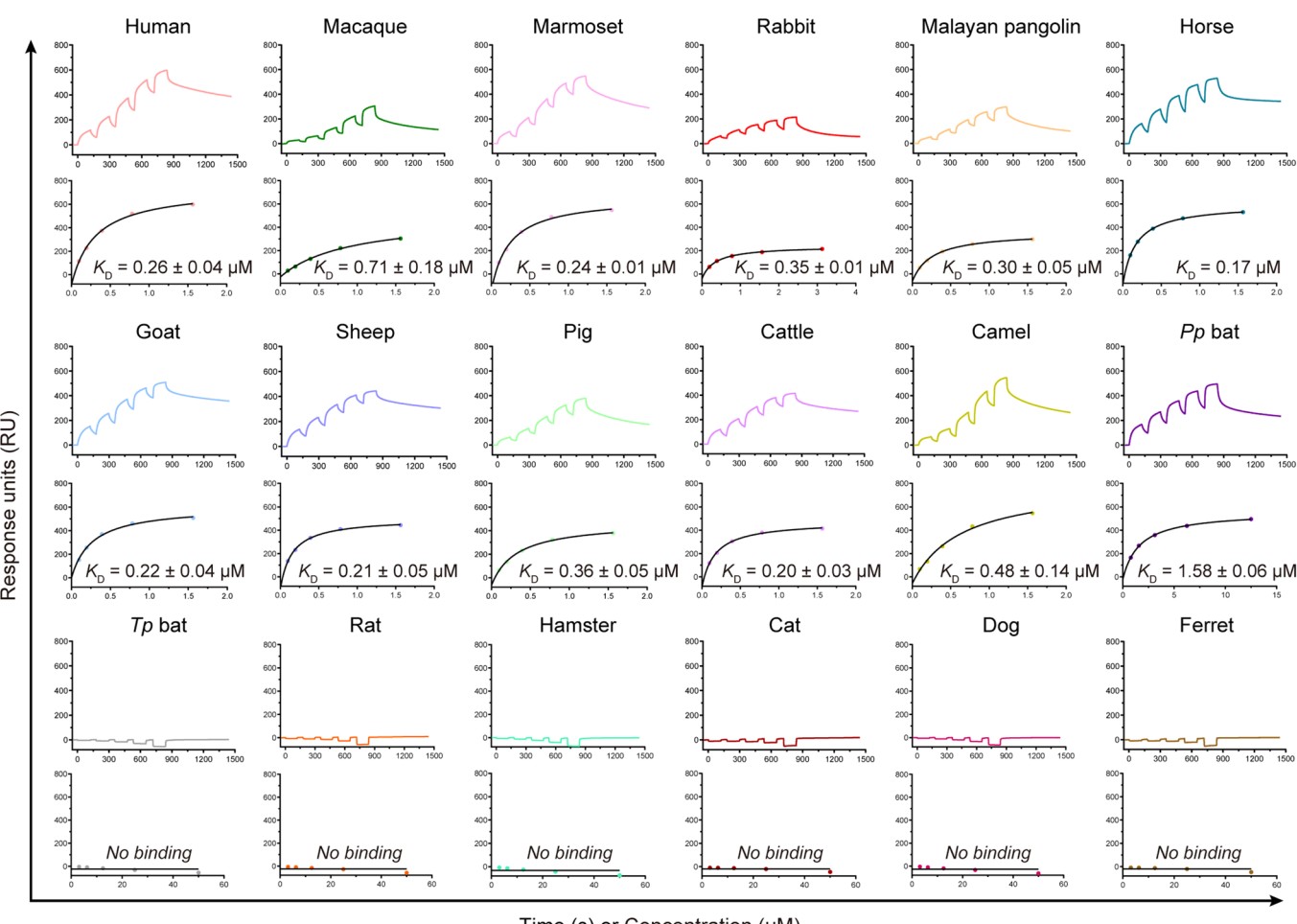

**Figure 5. The binding affinities between CD26 orthologs and the MjHKU4r-CoV-1 RBD.**

SPR analyses of binding between the MjHKU4r-CoV-1 RBD and 18 CD26 orthologs covering eight orders were performed. Raw curves are shown with different colors, as indicated. The fit curves are represented by black lines. $K_D$ values are the mean ± SD of three biological replicates. Source data are available online for this figure.

receptor. AA sequence alignment shows that these N-glycans, located at N229 on hCD26, are conserved among animal species. Our previous work showed that HKU4-CoV, utilizing the bat CD26, also needs N-glycan binding from the bat CD26 (Yuan et al, 2020). We assessed the mutations of N229 to A229 or Q229 to evaluate the N-glycans effect on the binding affinity. Unfortunately, hCD26 mutant proteins harboring A229 or Q229 were not successfully expressed, implying that the loss of N-glycans of N229 may adversely affect the expression, folding, or stability of hCD26.

The $K_D$ values for MjHKU4r-CoV-1 RBD binding to hCD26 reported in Fig. 1 (0.41 ± 0.03 μM), 2 (0.36 ± 0.04 μM), 5 (0.26 ± 0.04 μM), and 6 (0.23 ± 0.09 μM) are different, which may be caused by variability inherent to protein activity differences among different protein preparations, protein quantification by the bicinchoninic acid assay, and SPR measurements. Furthermore, the SPR results for the MjHKU4r-CoV-1, HKU4-CoV, and MERS-CoV binding to hCD26 displayed some differences from previous studies. The binding affinity between the MjHKU4r-CoV-1 RBD and hCD26 that we measured (0.41 μM) is higher than that of a

previous study (3.25 μM), as shown for the binding affinity between the HKU4-CoV RBD and hCD26 (5.62 μM vs. very weak) (Chen et al, 2023). In other reports, the affinity of the HKU4-CoV RBD for hCD26 is ~35.7 μM (Wang et al, 2014), which could be caused by the differences in measurement methods (SPR vs. biolayer interferometry, amine coupling vs. protein biotinylation, and CD26 immobilized and RBD flowed over vs. RBD immobilized and the CD26 flowed over). Given that CD26 was purified as a homodimer and RBD was purified as a monomer, the mode in which CD26 acts as the ligand and RBD as an analyte flowed over would be more suitable for 1:1 binding than the opposite way. Furthermore, to exclude potential interference with the functionally relevant part of the interacting molecule caused by amine coupling or protein biotinylation, a Twin-Strep-tag Capture Kit was used to capture hCD26 harboring a Twin-Strep tag at its C-terminus, and then the MjHKU4r-CoV-1 RBD with a His tag at its C-terminus was flowed over the sensor chip. The SPR results showed that the binding affinity between hCD26 and the MjHKU4r-CoV-1 RBD measured by this capture method (Fig. EV4) is similar to that measured from the amino coupling (Figs. 1B, 2A, 5 and 6B).

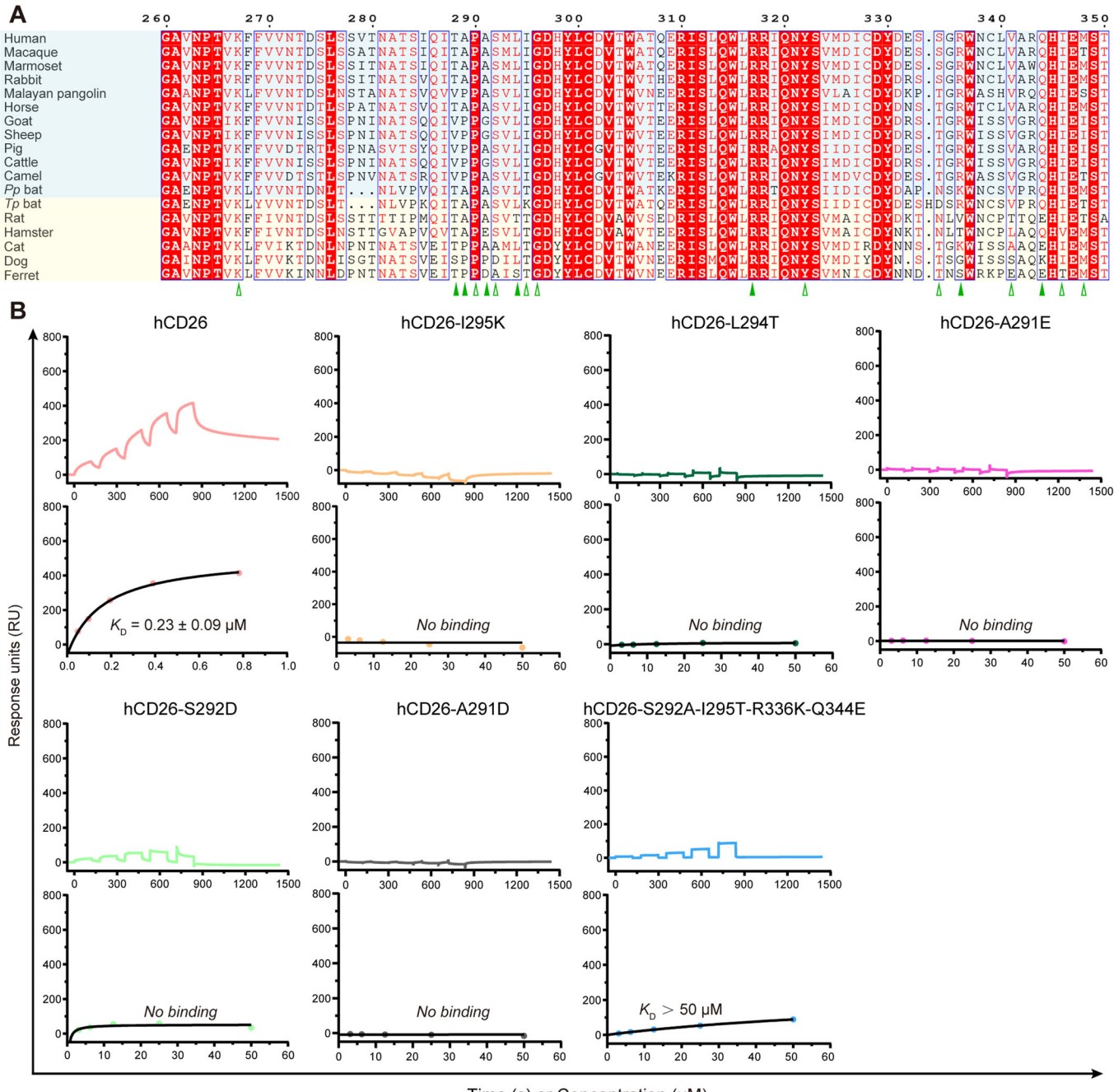

**Figure 6. Identification of host range determinants restricting MjHKU4r-CoV-1 recognition.**

(A) Part of the AA sequence alignment of the 18 CD26 orthologs from eight orders. Species that bind to the MjHKU4r-CoV-1 RBD are shaded in blue, otherwise in yellow. Residue sites on hCD26 interacting with the MjHKU4r-CoV-1 RBD are labeled using green triangles, and solid ones represent the residues participating in H-bonds. (B) SPR analysis of binding between the MjHKU4r-CoV-1 RBD and hCD26 and its seven mutants: hCD26-I295K, L294T, A291E, S292D, A291D, and S292A-I295T-R336K-Q344E. Raw curves are shown with different colors, as indicated. The fit curves are represented by black lines. $K_D$ values are the mean ± SD of three biological replicates. Source data are available online for this figure.

In our results, the binding affinity of the MjHKU4r-CoV-1 RBD for Malayan pangolin CD26 (0.30 ± 0.05 μM) was similar to that of hCD26 (0.26 ± 0.04 μM) (Fig. 5). AA sequence alignment for CD26s shows that Malayan pangolin CD26 has four AA distinctions participating in the interaction with the MjHKU4r-

CoV-1 RBD compared to hCD26, which were V288, P289, T334, and S348 (Fig. 6A). Structural homology modeling showed that these AAs could not significantly influence the binding of CD26 to the hydrophobic pocket or the strong polar interactions between the MjHKU4r-CoV-1 RBD and CD26 interface (Fig. 7A,B). In

**Table 3.   Key residue analysis for CD26s determining the host range of MjHKU4r-CoV-1.**

| Human | Tp bat | Rat | Hamster | Cat | Dog | Ferret |
|---|---|---|---|---|---|---|
| N229 | N229 | N229 | N229 | N229 | N229 | N229 |
| K267 | K267 | K267 | K267 | K267 | K267 | K267 |
| T288 | T288 | T288 | T288 | T288 | S288 (0.24 ± 0.07 µM) | T288 |
| A289 | A289 | A289 | A289 | P289 (0.30 ± 0.05 µM) | P289 (0.30 ± 0.05 µM) | P289 (0.30 ± 0.05 µM) |
| P290 | P290 | P290 | P290 | P290 | P290 | P290 |
| A291 | A291 | A291 | E291 (No binding) | A291 | P291 (0.38 ± 0.05 µM) | D291 (No binding) |
| S292 | S292 | S292 | S292 | A292 (0.38 ± 0.07 µM) | D292 (No binding) | A292 (0.38 ± 0.07 µM) |
| L294 | L294 | T294 (No binding) | L294 | L294 | L294 | S294 (11.03 ± 0.81 µM) |
| I295 | K295 (No binding) | T295 (0.62 ± 0.05 µM) | T295 (0.62 ± 0.05 µM) | T295 (0.62 ± 0.05 µM) | T295 (0.62 ± 0.05 µM) | T295 (0.62 ± 0.05 µM) |
| G296 | G296 | G296 | G296 | G296 | G296 | G296 |
| R317 | R317 | R317 | R317 | R317 | R317 | R317 |
| Y322 | Y322 | Y322 | Y322 | Y322 | Y322 | Y322 |
| S334 | D334 (0.50 ± 0.03 µM) | N334 (0.17 ± 0.01 µM) | N334 (0.17 ± 0.01 µM) | T334 (0.36 ± 0.02 µM) | T334 (0.36 ± 0.02 µM) | T334 (0.36 ± 0.02 µM) |
| R336 | R336 | V336 (0.47 ± 0.08 µM) | T336 (0.47 ± 0.03 µM) | K336 (0.48 ± 0.02 µM) | G336 (0.38 ± 0.02 µM) | S336 (0.57 ± 0.05 µM) |
| V341 | V341 | T341 (0.18 ± 0.02 µM) | L341 (0.14 ± 0.08 µM) | A341 (0.21 ± 0.01 µM) | S341 (0.19 ± 0.03 µM) | E341 (0.55 ± 0.01 µM) |
| Q344 | Q344 | E344 (0.33 ± 0.03 µM) | Q344 | E344 (0.33 ± 0.03 µM) | K344 (0.25 ± 0.04 µM) | E344 (0.33 ± 0.03 µM) |
| I346 | I346 | I346 | V346 (0.47 ± 0.02 µM) | I346 | I346 | T346 (0.20 µM) |
| M348 | T348 (0.37 ± 0.04 µM) | T348 (0.37 ± 0.04 µM) | M348 | M348 | M348 | M348 |
| I295-R336 | – | T295-V336 (36.03 ± 10.28 µM) | T295-T336 (23.73 ± 1.96 µM) | T295-K336 (10.87 ± 0.67 µM) | T295-G336 (14.65 ± 6.15 µM) | T295-S336 (20.30 ± 4.98 µM) |
| S292-I295-R336 | – | – | – | A292-T295-K336 (17.10 ± 0.82 µM) | – | – |
| S292-I295-R336-Q344 | – | – | – | A292-T295-K336-E344 (>50 µM) | – | – |

The AA sequences of animal CD26s were renumbered based on hCD26 for convenient comparison. Interacting residues of hCD26 with the MjHKU4r-CoV-1 RBD are listed in the first column. The corresponding AAs in the CD26s from Tp bat, rat, hamster, cat, dog, and ferret are listed. The $K_D$ value for hCD26 bound to the MjHKU4r-CoV-1 RBD was 0.23 ± 0.09 µM, and that for each mutant is filled in in parentheses. $K_D$ values shown are the mean ± SD of at least three biological replicates.

addition, the binding capacity of the MjHKU4r-CoV-1 RBD for Pp bat CD26 (1.58 ± 0.06 µM) was lower than that of hCD26 (0.26 ± 0.04 µM) (Fig. 5). AA sequence alignment for CD26s showed that Pp bat CD26 has three AA distinctions participating in the interaction with the MjHKU4r-CoV-1 RBD compared to hCD26, which were T295, N334, and K336. SPR results demonstrated that T295 and K336 substitutions decrease binding affinity to the MjHKU4r-CoV-1 RBD (Fig. EV3). As observed in the cat CD26, T295 could influence the binding of CD26 to the hydrophobic pocket, and K336 may influence the H-bond with Y463 of the MjHKU4r-CoV-1 RBD (Fig. 7A,B).

Altogether, our study here confirms the strong binding of hCD26 with the MjHKU4r-CoV-1 RBD by biochemical and structural analyses. We also demonstrate that MjHKU4r-CoV-1 has a broad host range by binding affinity analysis, and we decipher the determinants of this host spectrum. All of these data point to the potential emergence of human/animal infections by MjHKU4r-CoV-1.

## Methods

### Gene cloning, protein production, and purification

The DNA sequences encoding the MjHKU4r-CoV-1 RBD (S residues E375-Y614, GWHBHAL01000000) and HKU4-CoV RBD

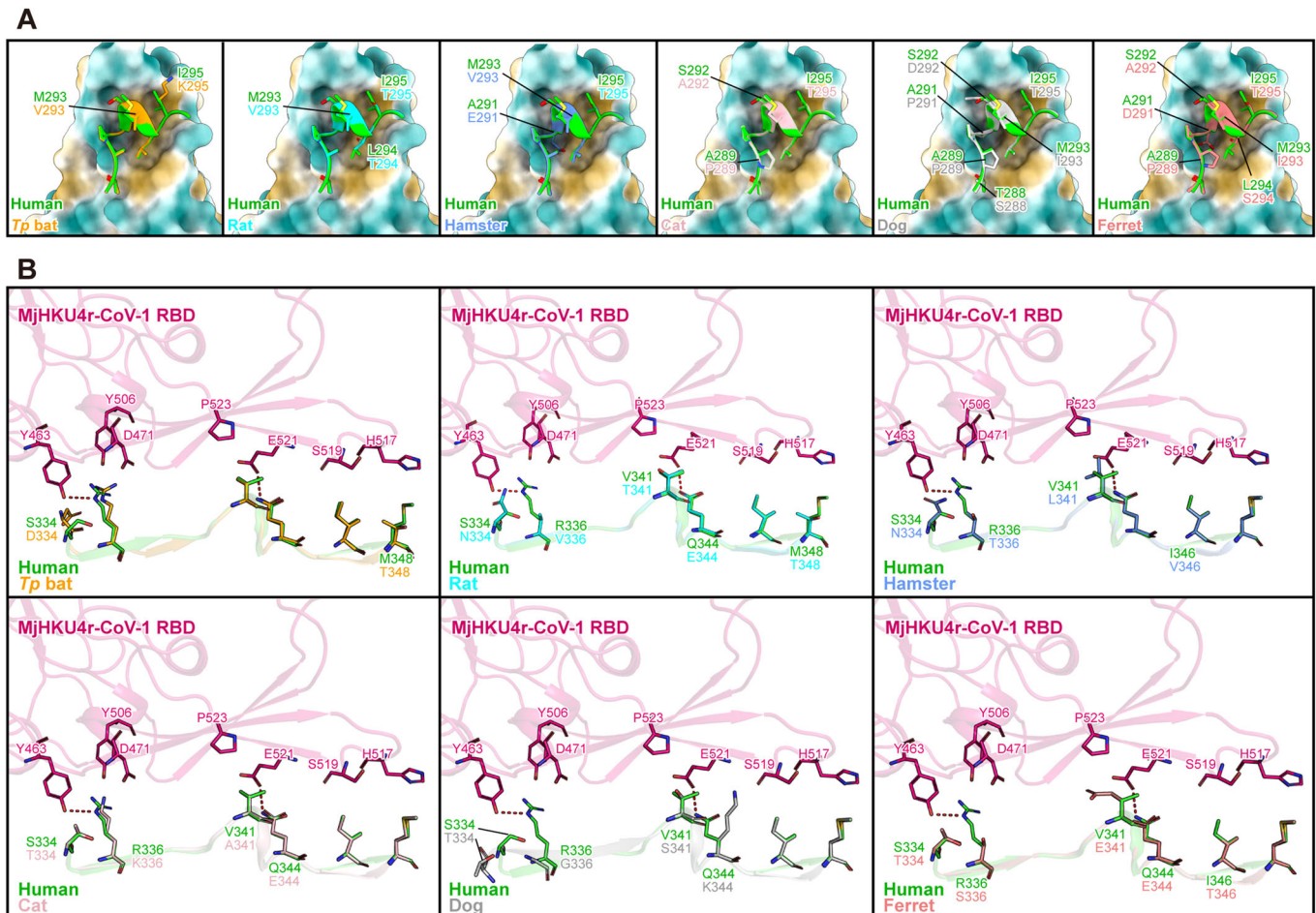

**Figure 7. Homology modeling and interface comparison.**

(A) Hydrophobic interaction interface comparisons between hCD26 and CD26s from *Tp* bat, rat, hamster, cat, dog, and ferret. These six animal CD26 structures from structural homology modeling are aligned with the MjHKU4r-CoV-1 RBD/hCD26 complex. The MjHKU4r-CoV-1 RBD is shown as a surface according to hydrophobicity (green: hydrophilic; white: neutral; and gold: hydrophobic). The helix (residues 289–296) from hCD26 and corresponding regions on the animal CD26s are shown as cartoons and sticks and colored in green (human), orange (*Tp* bat), cyan (rat), cornflower blue (hamster), pink (cat), light gray (dog), and light coral (ferret). Differential residues between the two CD26s for comparison are labeled in their respective colors. (B) Polar interaction comparisons between hCD26 and CD26s from *Tp* bat, rat, hamster, cat, dog, and ferret. Residues 334–348 from hCD26 and corresponding regions on the animal CD26s are shown as cartoons, which are colored corresponding to (A). Interacting residues in the MjHKU4r-CoV-1 RBD/hCD26 complex are shown as sticks, as well as corresponding residues from animal CD26s. Differential residues between hCD26 and animal CD26 are labeled in their respective colors.

(S residues E372-Y611, ABN10848.1) were separately cloned into a modified pFastbac Dual plasmid (Invitrogen) under the control of the polyhedrin promoter (Song et al, 2019). An N-terminal gp67 signal peptide and a C-terminal hexa-His tag were added to facilitate protein secretion and purification (Wang et al, 2014). Green fluorescent protein (GFP) was placed under the control of the P10 promoter to visualize its expression. Recombinant pFastbac Dual plasmid was transformed into DH10Bac-competent cells (Invitrogen, Cat# 10361-012) to produce the recombinant bacmid. Transfection and virus amplification were conducted with Sf9 cells (Invitrogen, Cat# 11496015), and the recombinant proteins were expressed in High Five cells (Invitrogen, Cat# B85502) for 2 days. Sf9 and High Five cells were cultured in Insect-XPRESS™ Medium (Lonza, Cat# 12-730Q) and SIM HF Medium (Sino Biological, Cat# MHF1), respectively. Cell supernatant was collected, and soluble

proteins were purified using HisTrap HP (Cytiva) and Superdex 200 Increase 10/300 GL (Cytiva) columns successively.

DNA sequences encoding the MERS-RBD (S residues E367-Y606, AFS88936.1), hCD26 (residues S39-P766, NP_001926.2), and seven hCD26 mutants (hCD26-K267A, R317A, R336A, T288A, Q344A, and N229A/Q) fused to a C-terminal 6 × His tag were separately cloned into the pCAGGS vector. A Kozak sequence and an exogenous signal peptide derived from μ-phosphatase (MGILPSPGM-PALLSLVSLLSVLLMGCVAETGT) were added to the N-terminus to maximize protein production (Zhao et al, 2023). HEK293F cells (Gibco, Cat# 11625-019) were used to express these proteins. Cells were cultured in SMM 293-TII medium (Sino Biological, Cat# M293TII) with 5% $CO_2$ at 37 °C and 140 rpm and then were transfected with transfection reagent (Sino Biological, Cat# STF02) at a density of $2 \times 10^6$ cells/mL. Expression Medium

**Table 4. Homology model assessment.**

| Species | Sequence coverage (%) | Sequence identity (%) | GMQE | QMEANDisCo Global | QSQE | RMSD |
|---|---|---|---|---|---|---|
| *Tp* bat | 100 | 83.01 | 0.92 | 0.86 ± 0.05 | 0.89 | 0.123 |
| Rat | 99 | 85.12 | 0.92 | 0.86 ± 0.05 | 0.85 | 0.131 |
| Hamster | 100 | 84.46 | 0.92 | 0.87 ± 0.05 | 0.86 | 0.124 |
| Cat | 100 | 88.03 | 0.94 | 0.88 ± 0.05 | 0.87 | 0.110 |
| Dog | 100 | 88.84 | 0.93 | 0.87 ± 0.05 | 0.79 | 0.108 |
| Ferret | 100 | 87.62 | 0.94 | 0.88 ± 0.05 | 0.78 | 0.115 |

Homology models derived from SWISS-Model for the CD26 protein from six species in complex with the MjHKU4r-CoV-1 RBD were assessed for sequence coverage and identity between each animal CD26 and hCD26, GMQE, QMEANDisCo Global, and QSQE. Each model was also aligned to the MjHKU4r-CoV-1 RBD/hCD26 complex to calculate the RMSD.

supplement (Sino Biological, Cat# M293-SUPI) was added at 24 h and 72 h after transfection. Cell supernatant was collected on the fifth day after transfection. Proteins were purified using HisTrap excel (Cytiva) and Superdex 200 Increase 10/300 GL (Cytiva) columns.

The DNA sequences encoding *Pp* bat CD26 (AGF80256.1), Malayan pangolin CD26 (XP_017519864.1), ferret CD26 (AIG55260.1), pig CD26 (NP_999422.1), goat CD26 (AIG55261.1), cat CD26 (NP_001009838.1), rabbit CD26 (XP_008256890.1), horse CD26 (XP_023478786.1), rat CD26 (NP_036921.2), macaque CD26 (NP_001034279.2), sheep CD26 (XP_004004709.1), hamster CD26 (NP_001297500.1), camel CD26 (AHK13386.1), marmoset CD26 (XP_002749438.1), cattle CD26 (NP_776464.1), dog CD26 (XP_038319326.1), *Tp* bat CD26 (AZO92860.1), and hCD26 (NP_001926.2), as well as 36 hCD26 mutants (hCD26-T288S, A289P, A291E/P/D, S292A/D, L294T/S, I295K/T, S334D/N/T, R336V/T/K/G/S, V341T/L/A/S/E, Q344E/K, I346V/T, M348T, I295T-R336K, I295T-R336V, I295T-R336T, I295T-R336G, I295T-R336S, S292A-I295T-R336K, and S292A-I295T-R336K-Q344E) fused to a C-terminal Twin-Strep tag, were separately cloned into the pCAGGS vector. A Kozak sequence and an exogenous signal peptide derived from μ-phosphatase (MGILPSPGM-PALLSLVSLLSVLLMGCVAETGT) were added to the N-terminus to maximize protein production (Zhao et al, 2023). HEK293F cells were used to express these proteins, which were purified using StrepTrap XT (Cytiva) and Superdex 200 Increase 10/300 GL (Cytiva) columns.

## Crystallization, data collection, and structure determination

The MjHKU4r-CoV-1 RBD and hCD26, both of which had a C-terminal hexa-His tag, were mixed at a molar ratio of 2.5:1 and incubated on ice overnight. Crystallization trials were set up with commercial crystallization kits (Molecular Dimensions) using the sitting drop vapor diffusion method. Protein complex (1 μL at 15 mg/mL in 20 mM Tris-HCl and 150 mM NaCl, pH 8.0) was mixed with 1 μL reservoir solution. The resultant drop was then sealed and equilibrated against 100 μL reservoir solution at 18 °C. Diffractable complex crystals were obtained in 0.1 M Amino acids, 0.1 M buffer system 3 (Tris base; Bicine), and 30% (v/v) EDO_P8K, pH 8.5, at 18 °C. Crystals were flash-cooled in liquid nitrogen after a brief soaking in a reservoir solution with the addition of 17% (v/v) glycerol. The X-ray diffraction data were collected under cryogenic conditions (100 K) at Shanghai Synchrotron Radiation Facility (SSRF), beamline BL02U1. The data were indexed, integrated, and

scaled with XDS (Kabsch, 2010). The complex structure of the MjHKU4r-CoV-1 RBD/hCD26 was solved by the molecular replacement method using Phaser (Read, 2001) with the structure of HKU4-CoV RBD/hCD26 (PDB: 4QZV). The atomic models were completed with Coot v.0.9.8 (Emsley and Cowtan, 2004) and refined with phenix.refine in Phenix v.1.20.1 (Adams et al, 2010). The final statistics for data collection and structure refinement are represented in Table 1. PyMOL v.2.4 (https://pymol.org/2/) and UCSF ChimeraX v.1.3 (Goddard et al, 2018) were used to generate the structural figures.

## SPR assay

A BIAcore 8K (Cytiva) was used to measure the binding affinities between CD26 and the RBD with a CM5 sensor chip (Cytiva) at 25 °C in single-cycle mode. All proteins were stored in PBS with 0.005% Tween-20 for use. hCD26, hCD26 mutants, and animal CD26 orthologs were diluted using the immobilization buffer and then immobilized on individual channels of a CM5 sensor chip using an Amine Coupling Kit (Cytiva, Cat# BR100633). Serially diluted MjHKU4r-CoV-1, HKU4-CoV, or MERS-CoV RBDs were flowed through the chip. The CM5 sensor chip was regenerated using 5 mM NaOH three times. Furthermore, hCD26 with Twin-Strep tag at its C-terminus was surface captured via StrepTactin XT from the Twin-Strep-Tag Capture Kit (IBA GmbH, Cat# 2-4370-000) covalently immobilized on a CM5 chip, and the MjHKU4r-CoV-1 RBD was diluted and flowed over the sensor chip. SPR results were analyzed by steady-state and kinetics (1:1 binding). The data were obtained from three biological replicates and are presented as the mean ± standard deviation (SD). Biacore Insight Evaluation software v.3.0 (Cytiva) and GraphPad Prism 8 were used to analyze data and make some figures.

## Homology modeling

Structural homology modeling of the CD26s from *Tp* bat, rat, hamster, cat, dog, and ferret in complex with the MjHKU4r-CoV-1 RBD was used to explore why they could not bind to the MjHKU4r-CoV-1 RBD. This was performed by the SWISS-MODEL server (Waterhouse et al, 2018) with the MjHKU4r-CoV-1 RBD/hCD26 complex structure in this work as the template. The quality of the models was assessed in SWISS-Model for sequence coverage, sequence identity, GMQE, QMEANDisCo Global, and QSQE (Waterhouse et al, 2018; Damas et al, 2020).

The models were then imported into PyMOL v.2.4 (https://pymol.org/2/), and the RMSD was calculated between each model and the template structure. Interacting residues of the six animal CD26s distinct from that of hCD26 were labeled and presented visually with PyMOL v.2.4 (https://pymol.org/2/) and UCSF ChimeraX v.1.3 (Goddard et al, 2018).

## Data availability

The atomic structure coordinates were deposited in the RCSB Protein Data Bank (PDB) under the accession codes 8WKU.

The source data of this paper are collected in the following database record: biostudies:S-SCDT-10_1038-S44319-024-00169-8.

## Peer review information

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

## Acknowledgements

We are grateful to Mingxiong Tian (Southeast University) for protein preparations. We thank Pengcheng Han (Southeast University) and Kefang Liu (Institute of Microbiology, CAS) for revising the manuscript. We thank the staff of the BL02U1 beamline at the Shanghai Synchrotron Radiation Facility. This work was supported by the National Key R&D Program of China (2022YFC2303401, 2022YFF1203203 and 2021YFA1300803 to George F Gao), the Strategic Priority Research Program of the Chinese Academy of Sciences (XDB29010202 to George F Gao), the fellowship of China Postdoctoral Science Foundation (2023M743731 to Zhennan Zhao), Postdoctoral Fellowship Program of CPSF (GZB20230821 to Zhennan Zhao) and Young Elite Scientists Sponsorship Program by CAST (2023QNRC001 to Zhennan Zhao).

## Author contributions

**Zhennan Zhao**: Conceptualization; Resources; Data curation; Software; Formal analysis; Funding acquisition; Visualization; Methodology; Writing—original draft. **Xin Li**: Formal analysis; Methodology. **Yan Chai**: Software; Methodology. **Zhifeng Liu**: Methodology. **Qihui Wang**: Formal analysis; Writing—review and editing. **George F Gao**: Conceptualization; Formal analysis; Supervision; Funding acquisition; Writing—review and editing.

Source data underlying figure panels in this paper may have individual authorship assigned. Where available, figure panel/source data authorship is listed in the following database record: biostudies:S-SCDT-10_1038-S44319-024-00169-8.

## Disclosure and competing interests statement

The authors declare no competing interests.

# Expanded View Figures

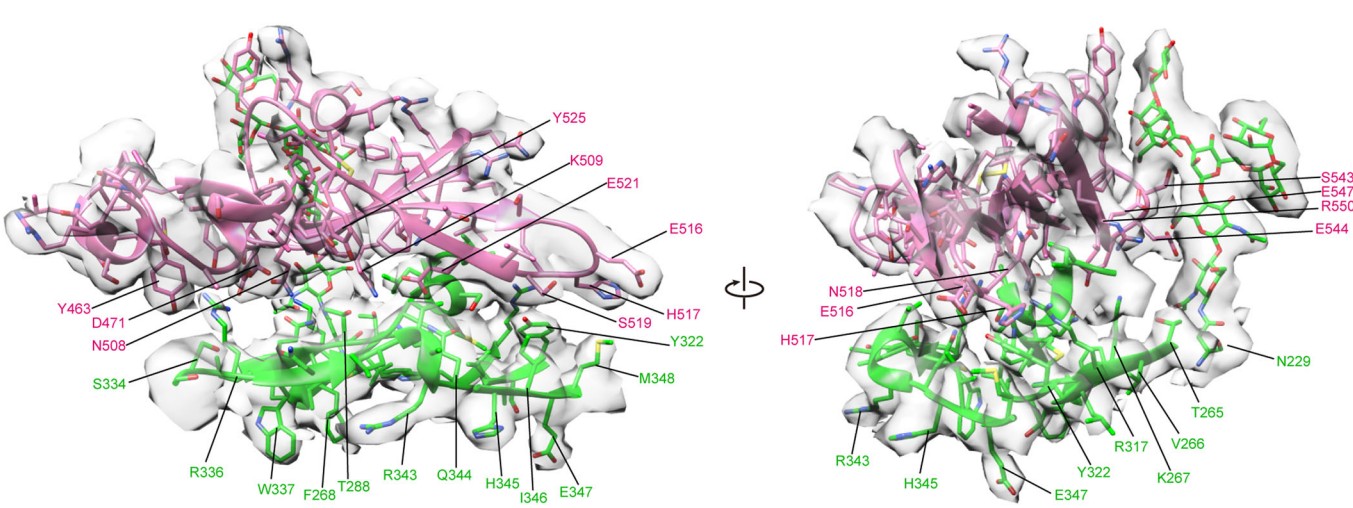

**Figure EV1.  Local density map of the interface between the MjHKU4r-CoV-1 RBD and hCD26.**

The MjHKU4r-CoV-1 RBD and hCD26 are colored in hot pink and green, respectively. The local 2Fo-Fc map contoured at 0.5σ for its binding interface is shown as gray surfaces, and AAs are displayed as sticks.

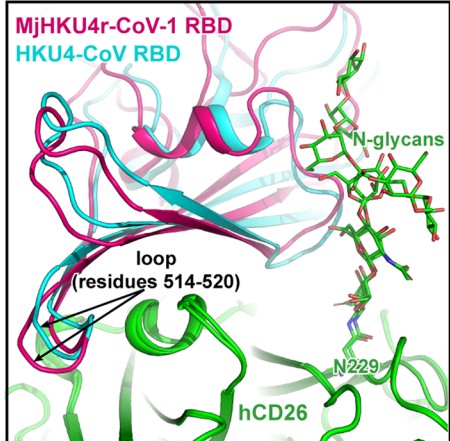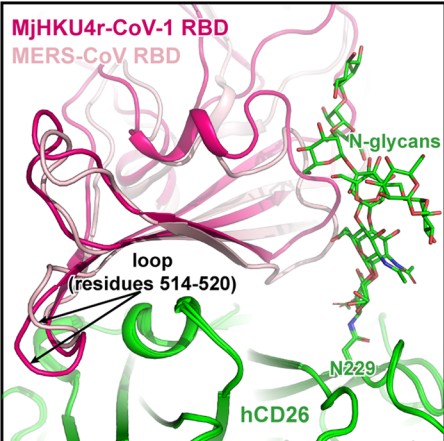

**Figure EV2.   Structural comparisons among the MjHKU4r-CoV-1 RBD/hCD26, HKU4-CoV RBD/hCD26, and MERS-CoV RBD/hCD26 complexes.**

The MjHKU4r-CoV-1, HKU4-CoV, and MERS-CoV RBDs and hCD26 are colored corresponding to Fig. 4.

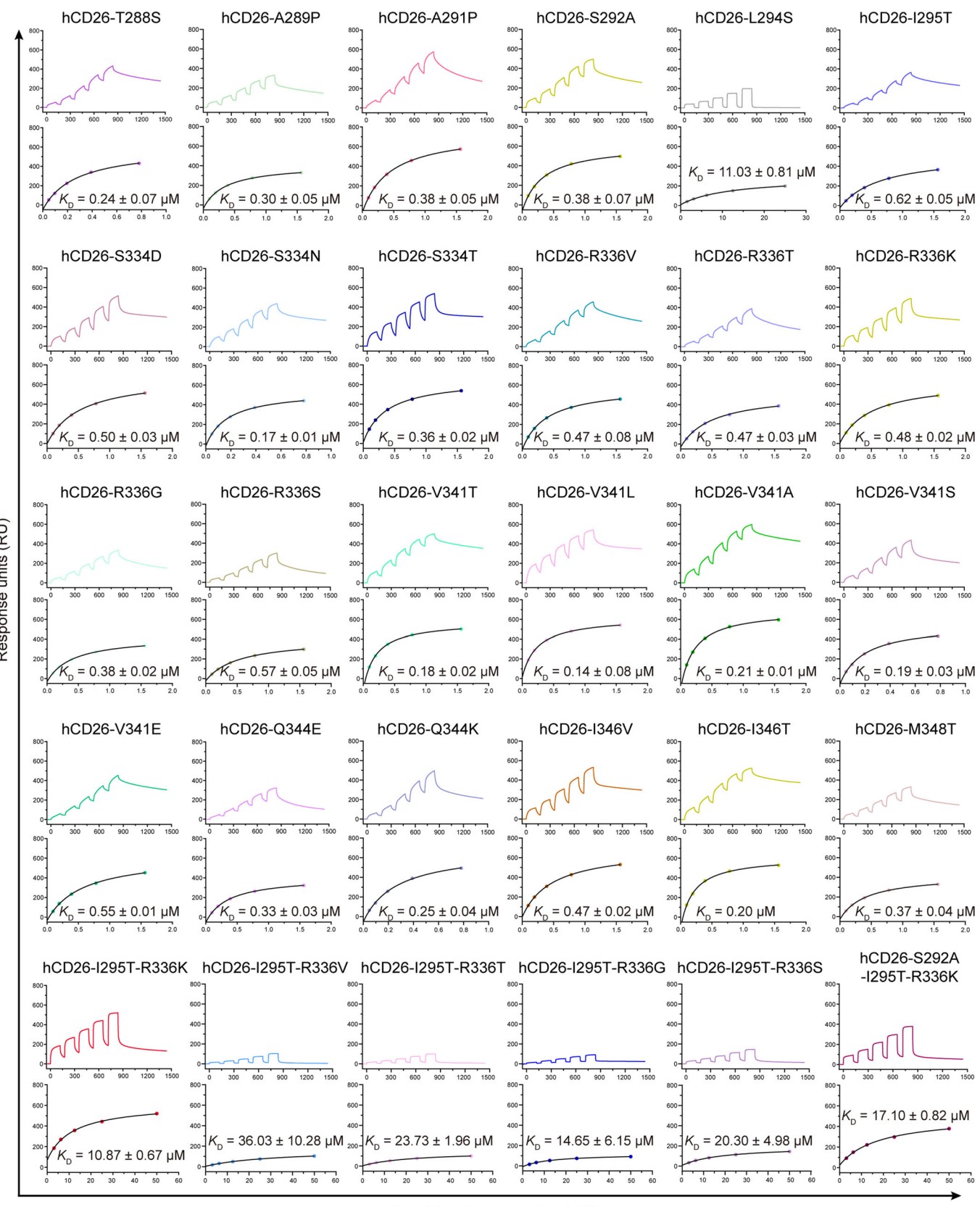

◀ **Figure EV3. Identification of crucial residue determinants for the host range of MjHKU4r-CoV-1.**

SPR analysis of binding between the MjHKU4r-CoV-1 RBD and hCD26 mutants: hCD26-T288S, A289P, A291P, S292A, L294S, I295T, S334D/N/T, R336V/T/K/G/S, V341T/L/A/S/E, Q344E/K, I346V/T, M348T, I295T-R336K, I295T-R336V, I295T-R336T, I295T-R336G, I295T-R336S, and S292A-I295T-R336K. Raw curves are shown as the indicated colors. The fit curves are represented by black lines. $K_D$ values are the mean ± SD of three biological replicates. Source data are available online for this figure.

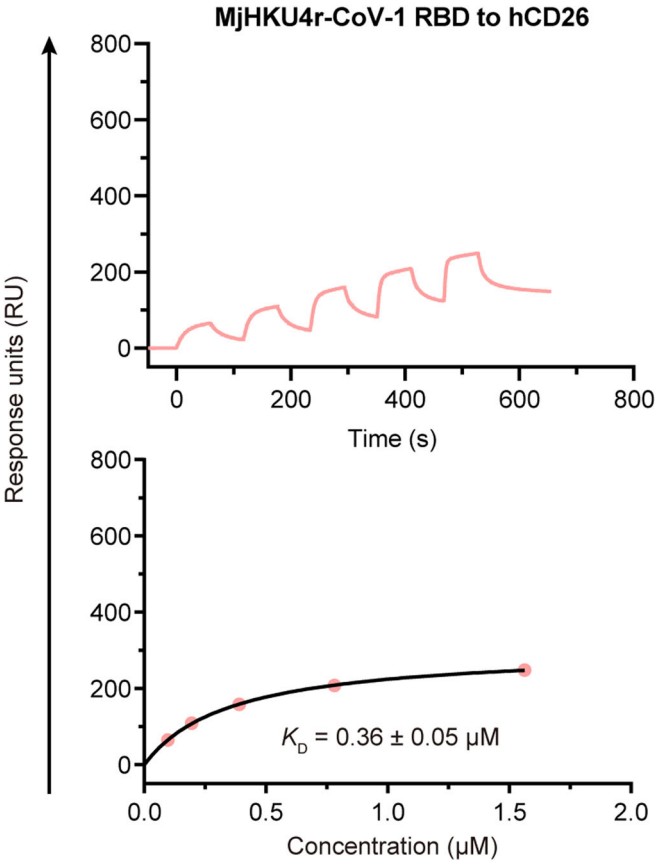

**Figure EV4. Binding affinity measurement between the MjHKU4r-CoV-1 RBD and hCD26 using a capture method.**

Raw curves are shown in pink, and the fit curves are represented by black lines. $K_D$ values are the mean ± SD of three biological replicates. Source data are available online for this figure.

