## [Peer Review File · EMBO Reports]

Molecular basis for receptor recognition and broad host tropism for merbecovirus MjHKU4r-CoV-1

Zhennan Zhao, Xin Li, Yan Chai, Zhifeng Liu, Qihui Wang, and George Gao

Corresponding author(s): George Gao (gaof@im.ac.cn)

Review Timeline:

Submission Date:	9th Jan 24
Editorial Decision:	10th Jan 24
Revision Received:	13th Apr 24
Editorial Decision:	17th May 24
Revision Received:	23rd May 24
Accepted:	24th May 24

Editor: Achim Breiling

Transaction Report: This manuscript was transferred to EMBO reports following peer review at The EMBO Journal.

Dear Prof. Gao,

Thank you for transferring your manuscript to EMBO reports. I now went through the manuscript and the referee reports from The EMBO Journal (attached again below). The referees have several concerns and suggestions to improve the manuscript, or to strengthen the data and the conclusions drawn.

Given the constructive referee comments, I would like to invite you to revise your manuscript with the understanding that all concerns of the referees must be addressed in the revised manuscript or in a finalized detailed point-by-point response. Acceptance of your manuscript will depend on a positive outcome of another round of review at EMBO reports, using the same referees.

EMBO reports emphasizes novel functional over detailed mechanistic insight, but asks for strong in vivo relevance of the findings, and clear experimental support of the major conclusions. Thus, we will not require addressing points regarding more mechanism experimentally. However, it will be necessary that during final revision you address all points questioning the main conclusions of the study, and all technical concerns, or points regarding the experimental designs, model systems used, or data presentation.

Please also have your revised manuscript carefully proofread by a native speaker (see also the reports of referees #2 and #3)

1) a .docx formatted version of the final manuscript text (including legends for main figures, EV figures and tables), but without the figures included. Please make sure that changes are highlighted to be clearly visible. Figure legends should be compiled at the end of the manuscript text.

2) individual production quality figure files as .eps, .tif, .jpg (one file per figure), of main figures and EV figures. Please upload these as separate, individual files upon re-submission. Please make sure that all figure panels are called out separately and sequentially in the manuscript text

For more details please refer to our guide to authors:

See also our guide for figure preparation:

Moreover, please consult our guidelines for figure legend preparation:

4) a complete author checklist, which you can download from our author guidelines

(<https://www.embopress.org/page/journal/14693178/authorguide>). Please insert page numbers in the checklist to indicate where

the requested information can be found in the manuscript. The completed author checklist will also be part of the RPF.

Please also follow our guidelines for the use of living organisms, and the respective reporting guidelines:
<http://www.embopress.org/page/journal/14693178/authorguide#livingorganisms>

5) that primary datasets produced in this study (e.g. RNA-seq, ChIP-seq and array data) are deposited in an appropriate public database. This is now mandatory (like the COI statement). If no primary datasets have been deposited in any database, please state this in this section (e.g. 'No primary datasets have been generated and deposited').

The accession numbers and database should be listed in a formal "Data Availability " section (placed after Materials & Methods) that follows the model below. Please note that the Data Availability Section is restricted to new primary data that are part of this study.

Data availability

8) Regarding data quantification and statistics, please make sure that the number "n" for how many independent experiments were performed, their nature (biological versus technical replicates), the bars and error bars (e.g. SEM, SD) and the test used to calculate p-values is indicated in the respective figure legends (also for potential EV figures and all those in the final Appendix). Please also check that all the p-values are explained in the legend, and that these fit to those shown in the figure. Please provide statistical testing where applicable. Please avoid the phrase 'independent experiment', but clearly state if these were biological or technical replicates. Please also indicate (e.g. with n.s.) if testing was performed, but the differences are not significant. In case n=2, please show the data as separate datapoints without error bars and statistics.

See also:

<http://www.embopress.org/page/journal/14693178/authorguide#statisticalanalysis>

9) Please note our reference format:

10) We updated our journal's competing interests policy in January 2022 and request authors to consider both actual and perceived competing interests. Please review the policy <https://www.embopress.org/competing-interests> and add a statement declaring your competing interests. Please name that section 'Disclosure and Competing Interests Statement' and add it after the author contributions section.

11) Please order the manuscript sections like this using these names:

Title page - Abstract - Keywords - Introduction - Results - Discussion - Materials and Methods - Data availability section (DAS) - Acknowledgements - Disclosure and Competing Interests Statement - References - Figure legends - Expanded View Figure

legends

12) Please make sure that all the funding information is also entered into the online submission system and is complete and similar to the one in the manuscript text file (in the Acknowledgements).

13) We now use CRediT to specify the contributions of each author in the journal submission system. CRediT replaces the author contribution section. Please use the free text box to provide more detailed descriptions. Thus, please do not provide your final manuscript text file with an author contributions section. See also guide to authors: <https://www.embopress.org/page/journal/14693178/authorguide#authorshipguidelines>

14) We would encourage you to use 'Structured Methods', our new Materials and Methods format. According to this format, the Materials and Methods section should include a Reagents and Tools Table (listing key reagents, experimental models, software and relevant equipment and including their sources and relevant identifiers) followed by a Methods and Protocols section in which we encourage the authors to describe their methods using a step-by-step protocol format with bullet points, to facilitate the adoption of the methodologies across labs. More information on how to adhere to this format as well as downloadable templates (.doc or .xls) for the Reagents and Tools Table can be found in our author guidelines (section 'Structured Methods'):

I look forward to seeing a revised version of your manuscript when it is ready. Please let me know if you have questions or comments regarding the revision.

Yours sincerely,

Referee #1:

Zhao et al describe a structural study of receptor recognition by a novel merbecovirus, MjHKU4r-CoV-1, which is related to HKU-4-CoV. The study is an extension of a previously published study (Chen et al, Cell 186:850-863). Chen et al showed that MjHKU4r-CoV-1 has a broad host tropism. The present study describes the basis for this broad tropism, using RBD/hDPP4 crystallization, SPR binding and molecular modeling. The results are clearly written and are important to the field. The study is entirely structural, but I think would be of interest to a broad audience of EMBO J readers since the virus would be an example of a merbecovirus that could cross species to infect humans. Please note that this reviewer is not a structural biologist, so cannot comment on the technical aspects of the paper.

Specific comment:

The study is limited to RBD binding. If there is any information available about whether other parts of the S or S1 protein, particularly the NTD, affect hDPP4 binding, this information should be included.

Referee #2:

Zhao et al. report structural and biophysical analyses of the RBD of recently identified pangolin merbecovirus MjHKU4r-CoV-1 (reported by Chen et al. in February 2023, PMID: 36803605), which utilizes cellular receptor CD26 for entry, similar to MERS-CoV and a bat virus HKU4-CoV. They report an X-ray structure of MjHKU4r-CoV-1 RBD bound to human CD26, and characterization of the RBD interactions with CD26 from a wide range of mammals, using mutational analyses of the human CD26 and SPR measurements. Detailed structural comparisons of the MjHKU4r-CoV-1 bound to human CD26 with the CD26 complexes with HKU4-CoV and MERS-CoV RBD (structures previously reported) provide some indications into residues that may be important molecular determinants.

The main conclusions of the manuscript, that MjHKU4r-CoV-1 RBD binds to human CD26 with high affinity, and that it has a broad host range, are unfortunately not novel and were reported by Chen et al. earlier this year (PMID: 36803). The structural comparative analyses, while carefully done, provide only ideas as to which residues could be mutated to obtain convincing functional data into the molecular determinants of the MjHKU4r-CoV-1 RBD interactions with CD26. These binding experiments (or a functional assay substitute) have not been done. Thus, the manuscript fails short of having a broad biological significance.

Some major points to improve manuscript:

- 1) Bulk of the structural comparisons and conclusions are based on the binding affinities of the RBDs and human CD26. The measurements were done using SPR, with CD26 that was immobilized to the sensor chips via amine links.
 - a. There are several incongruences between the calculated KD values here and reported by Chen et al. For example, the KD value for MjHKU4r-CoV-1 RBD binding to human CD26 is estimated to $0.8 \pm 0.1 \mu\text{M}$ (Figure 1B), and the value reported by Chen et al. is $3.25 \mu\text{M}$. The binding to the *Tylonycteris pachypus* bat CD26 could not be observed here (Figure 5), while KD similar to that of the binding to human CD26 was reported by Chen et al. The authors mention briefly (line 269) that the discrepancies could be due to different techniques used (SPR vs BLI used by Chen et al.). This claim would be strengthened by additional experiments, for example performing SPR the reverse way i.e. with the RBD immobilized and the receptor flow over, or producing either RBD or CD26 with a tag that allows specific labelling or immobilization without potentially interfering with the functionally relevant part of the interacting molecule (which is the case with amine coupling or protein biotinylation)
 - b. Different KD values for MjHKU4r-CoV-1 RBD binding to human CD26 are reported on figures 1 ($0.8 \mu\text{M}$), 2 ($0.36 \mu\text{M}$) and 5 ($0.26 \mu\text{M}$). Could the authors comment if this is an expected variability inherent to SPR measurements, or there might be another reason behind it.
- 2) Homology modelling of CD26 from bat, rat, hamster etc (Figure 6) was performed using SWISS-MODEL server. Does this prediction program provide any statistical parameters that would attest to the accuracy of the predicted models? Without these values, the analyses that ensue (lines 218-245) remain weak. Could do the authors perform the same analyses with the AlphaFold2 multimer program, which gives clear confidence parameters (PMID: 35365655)?
- 3) The main point of the paper, i.e. which residues determine the host range (lines 212-245), would benefit from being more succinct and clearer. This reviewer was lost looking for exactly which residues would be good candidates for mutagenesis and further studies, after reading the section couple of times. Having a figure that summarizes the residues the authors think would be important would be helpful.

Some minor points:

- 1) English language use needs improvement. A native speaker or an English language editor should work on the manuscript.

Referee #3:

Earlier in 2023, the group of Zengli Shi reported a new MERS-CoV-like coronavirus isolated from pangolins. The authors of the present study expressed the receptor-binding domain (RBD) of the pangolin MERS-related CoV (MjHKU4r-CoV-1) and studied its interaction with the receptor, human CD26 (hCD26; DPP4). The new virus is related to the betacoronaviruses of clade C, HKU4 and MERS-CoV, and there are indications that it can infect human cells more efficiently than bat CoVs.

A central result of this manuscript is the crystal structure of the complex between the RBD of the S protein of MjHKU4r and human CD26.

The authors used surface plasmon resonance (SPR) to determine the binding affinities of the MjHKU4r, the HKU4, and the MERS-CoV RBDs to the CD26 of 18 different animal species (including human and other primates, Malayan pangolin, various bat species etc). Site-directed mutagenesis was used to assess the role of individual residues of hCD26, including that of the glycan attached to Asn229.

To determine the host range of the new virus, the authors also made use of homology modeling. The biophysical data revealed that the Kd of the MjHKU4r and MERS-CoV RBDs to hCD26 is about equal and approx. 8 times lower than that of HKU4. This is in spite of the finding that among the three compared structures, MjHKU4r has the smallest interaction surface with hCD26 (Line 150).

The manuscript offers a very detailed description of the atomic interactions between the three RBDs on the one hand and hCD26 on the other. This is appreciated because it is necessary for understanding the differences in host range of the three viruses although I admit that when reading, I had a tendency to get tired with so much detail... However, Table 2 helps to counteract this effect.

In summary, the work of Zhao et al. demonstrates that the pangolin HKU4-like virus MjHKU4r-CoV-1 binds to human CD26 with relatively high affinity. Together with the broad host range found for this virus, it is conceivable that MjHKU4r-CoV-1 has the potential to be transmitted to humans, especially in view of the fact that pangolins are the most trafficked mammals. Therefore, this manuscript makes an important contribution to getting prepared for a potential MjHKU4r spillover into the human population. It also furthers our knowledge of the details of virus - CD26 interaction.

I have some individual comments:

Lines 25 - 26: Wang et al. (2014) is not the correct reference here. The correct reference is P.C.Y. Woo et al. (2012), which is also in the list of references. This should be corrected. Also, the authors should not state that coronaviruses "could be classified" into alpha, beta, gamma, and delta coronaviruses, as the scientific consensus is to classify them this way. Replace "could be" by "are"...

Lines 26 - 27: This statement is not true for clade A of betacoronaviruses. Only betacoronaviruses of clades B and C can be considered "life-threatening".

Line 27: There have not been "many epidemics and pandemics" caused by CoVs "in the past half-century". A little more accuracy in the description of the history of CoVs would be advisable. The first epidemic caused by a CoV was SARS-CoV in 2003, followed by MERS-CoV in 2012 (and extending over many years), and SARS-CoV-2 (since 2020, and still ongoing). The latter caused the only pandemic among the three. So beta-CoV outbreaks have been recorded since twenty years, not over "half a century".

Line 120: "In the MjHKU4r-CoV-1 RBD/hACE2 complex": Replace "hACE2" by "hCD26".

Line 226: K is the one-letter symbol for lysine, not arginine.

Line 232: "... many VDWs...": This is jargon and should be defined upon first occurrence.

The manuscript is written in a reasonable English, although at many places, articles are missing or have been used in the wrong way. There is certainly room for improvement on the language side.

Referee #1:

Zhao et al describe a structural study of receptor recognition by a novel merbecovirus, MjHKU4r-CoV-1, which is related to HKU-4-CoV. The study is an extension of a previously published study (Chen et al, Cell 186:850-863). Chen et al showed that MjHKU4r-CoV-1 has a broad host tropism. The present study describes the basis for this broad tropism, using RBD/hDPP4 crystallization, SPR binding and molecular modeling. The results are clearly written and are important to the field.

The study is entirely structural, but I think would be of interest to a broad audience of EMBO J readers since the virus would be an example of a merbecovirus that could cross species to infect humans. Please note that this reviewer is not a structural biologist, so cannot comment on the technical aspects of the paper.

Specific comment:

The study is limited to RBD binding. If there is any information available about whether other parts of the S or S1 protein, particularly the NTD, affect hDPP4 binding, this information should be included.

Response: Thanks for the suggestions. Previous studies demonstrate that CD26 interacts with the RBD of MERS-CoV, not the NTD (Guangwen Lu et al., 2013, Nature; Yuan Yuan et

al., 2020, J. Virol.). MjHKU4r-CoV-1 is a close relative of MERS-CoV and shares 66.23% amino acid sequence identity with MERS-CoV in their S proteins. Recently reported MjHKU4r-CoV-1 also uses its RBD to interact with CD26 for virus entry (Jing Chen et al., 2023, Cell). This information has been included in the introduction section (lines 68-70).

Referee #2:

Zhao et al. report structural and biophysical analyses of the RBD of recently identified pangolin merbecovirus MjHKU4r-CoV-1 (reported by Chen et al. in February 2023, PMID: 36803605), which utilizes cellular receptor CD26 for entry, similar to MERS-CoV and a bat virus HKU4-CoV. They report an X-ray structure of MjHKU4r-CoV-1 RBD bound to human CD26, and characterization of the RBD interactions with CD26 from a wide range of mammals, using mutational analyses of the human CD26 and SPR measurements. Detailed structural comparisons of the MjHKU4r-CoV-1 bound to human CD26 with the CD26 complexes with HKU4-CoV and MERS-CoV RBD (structures previously reported) provide some indications into residues that may be important molecular determinants.

The main conclusions of the manuscript, that MjHKU4r-CoV-1 RBD binds to human CD26 with high affinity, and that it has a broad host range, are unfortunately not novel and were reported by Chen et al. earlier this year (PMID: 36803). The structural comparative analyses, while carefully done, provide only ideas as to which residues could be mutated to obtain convincing functional data into the molecular determinants of the MjHKU4r-CoV-1 RBD interactions with CD26. These binding experiments (or a functional assay substitute) have not been done. Thus, the manuscript fails short of having a broad biological significance.

Response: Thanks for the comments. In the revised paper, these binding experiments have been done by site-directed mutagenesis and relative binding.

Some major points to improve manuscript:

1) Bulk of the structural comparisons and conclusions are based on the binding affinities of the RBDs and human CD26. The measurements were done using SPR, with CD26 that was immobilized to the sensor chips via amine links.

a. There are several incongruences between the calculated K_D values here and reported by Chen et al. For example, the K_D value for MjHKU4r-CoV-1 RBD binding to human CD26 is estimated to $0.8 \pm 0.1 \mu\text{M}$ (Figure 1B), and the value reported by Chen et al. is $3.25 \mu\text{M}$. The binding to the *Tylosycteris pachypus* bat CD26 could not be observed here (Figure 5), while K_D similar to that of the binding to human CD26 was reported by Chen et al. The authors mention briefly (line 269) that the discrepancies could be due to different techniques used (SPR vs BLI used by Chen et al.). This claim would be strengthened by additional experiments, for example performing SPR the reverse way i.e. with the RBD immobilized and the receptor flown over, or producing either RBD or CD26 with a tag that allows specific labelling or immobilization without potentially interfering with the functionally relevant part of the interacting molecule (which is the case with amine coupling or protein biotinylation)

Response: Thanks for the suggestions. Given that CD26 was purified as a homodimer and RBD was purified as a monomer, the mode in which CD26 acts as the ligand and RBD as an analyte flowed over would be more suitable for 1:1 binding than the opposite way. Furthermore, to exclude potential interference with the functionally relevant part of the

interacting molecule caused by amine coupling or protein biotinylation, a Twin-Strep-tag Capture Kit (IBA GmbH, Cat# 2-4370-000) was used to capture hCD26 harboring a Twin-Strep tag at its C-terminus, and then the MjHKU4r-CoV-1 RBD with a His tag at its C-terminus was flowed over the sensor chip. The SPR results showed that the binding affinity between hCD26 and the MjHKU4r-CoV-1 RBD measured by this capture method (Fig EV3) is similar to that measured from the amino coupling (Figs 1B, 2A, 5 and 6B). This part has been discussed in the discussion section (lines 306-322).

b. Different KD values for MjHKU4r-CoV-1 RBD binding to human CD26 are reported on figures 1 (0.8 μ M), 2 (0.36 μ M) and 5 (0.26 μ M). Could the authors comment if this is an expected variability inherent to SPR measurements, or there might be another reason behind it.

Response: Thanks for the suggestions. This difference may be caused by variability inherent to protein activity differences among batches, protein quantification by the bicinchoninic acid assay and SPR measurements, which has been discussed in the discussion section (lines 302-306). Furthermore, we also repeated the SPR assay to measure the binding affinity between hCD26 and MjHKU4r-CoV-1, HKU4 and MERS-CoV RBDs, which showed similar affinities to that previously reported in this paper, and Fig 1B has been updated.

2) Homology modelling of CD26 from bat, rat, hamster etc (Figure 6) was performed using SWISS-MODEL server. Does this prediction program provide any statistical parameters that

would attest to the accuracy of the predicted models? Without these values, the analyses that ensue (lines 218-245) remain weak. Could do the authors perform the same analyses with the AlphaFold2 multimer program, which gives clear confidence parameters (PMID: 35365655)?

Response: Thanks for the comments. SWISS-MODEL was widely used to build protein homology models from templates with high sequence similarities (Panke Qu et al., 2024, Cell; Jia Duan et al., 2023, Nat. Commun.; Gabriele Cerutti et al., 2021, Cell Host Microbe; Guangjin Xun et al., 2021, J. Virol.; Joana Damas et al., 2020, Proc. Natl. Acad. Sci. U. S. A.; Wei Liu et al., 2020, Nat. Microbiol; Ze Li et al., 2020, Nat. Struct. Mol. Biol.). The server provides several statistical parameters to evaluate the accuracy of the predicted models, including GMQE, QMEANDisCo Global, and QSQE. GMQE and QMEANDisCo Global give an overall model quality measurement between 0 and 1, with higher numbers indicating higher expected quality. GMQE is coverage dependent, i.e., a model covering only half of the target sequence is unlikely to get a score above 0.5. QMEANDisCo Global evaluates the model 'as is' without explicit coverage dependency, and its score of a model must be greater than 0.5. The QSQE score is between 0 and 1, reflecting the expected accuracy of the interchain contacts for a model built based on a given alignment and template. Generally, a higher QSQE is "better, " while a value above 0.7 can be considered reliable to follow the predicted quaternary structure in the modeling process. QSQE score complements the GMQE score which estimates the accuracy of the tertiary structure of the resulting model. Table 4 in the revised paper provides the parameters for evaluating the predicted models' accuracy.

The AlphaFold2 multimer program was also used to predict the MjHKU4r-CoV-1 RBD-CD26 complex for these six animals, as shown below.

Structural comparison with MjHKU4r-CoV-1 RBD-hCD26 complex for six complex

models. Six animal CD26s structures from *Tylonycteris pachypus* bat (orange, A), rat (cyan, B), hamster (cornflower blue, C), cat (pink, D), dog (light gray, E) and ferret (light coral, F) in complex with MjHKU4r-CoV-1 RBD (hot pink) were generated using the AlphaFold2 multimer program. Then, they were aligned with the MjHKU4r-CoV-1 RBD/hCD26 structure (black) determined by this paper. The iptm+ptm score and RMSD for each model were labeled.

Only *Tylonycteris pachypus* bat CD26 was predicted to form a similar interface to the hCD26; the other five animal CD26s were modeled to use distinct interfaces to bind MjHKU4r-CoV-1 RBD. Given that the AA sequence of these six animal CD26s has 83.01% to 88.84% identities to that of the hCD26 (Table 4), it is feasible that there is a lack of major conformational changes between species and a similar binding interface to hCD26. Thus, the

Swiss-model program would be more suitable for this analysis than the AlphaFold2 multimer program.

3) The main point of the paper, i.e. which residues determine the host range (lines 212-245), would benefit from being more succinct and clearer. This reviewer was lost looking for exactly which residues would be good candidates for mutagenesis and further studies, after reading the section couple of times. Having a figure that summarizes the residues the authors think would be important would be helpful.

Response: Thanks for the suggestions. This section has been rewritten, and Table 3 has been provided for clear illustration. Furthermore, single-site and multiple-site mutagenesis analysis was performed to evaluate the determinant residues for the host range (Figs 6B and EV2).

Some minor points:

1) English language use needs improvement. A native speaker or an English language editor should work on the manuscript.

Response: Thanks for the suggestions. This manuscript has been polished by a native speaker.

Referee #3:

Earlier in 2023, the group of Zengli Shi reported a new MERS-CoV-like coronavirus isolated from pangolins. The authors of the present study expressed the receptor-binding domain (RBD) of the pangolin MERS-related CoV (MjHKU4r-CoV-1) and studied its interaction with the receptor, human CD26 (hCD26; DPP4). The new virus is related to the betacoronaviruses of clade C, HKU4 and MERS-CoV, and there are indications that it can infect human cells more efficiently than bat CoVs.

A central result of this manuscript is the crystal structure of the complex between the RBD of the S protein of MjHKU4r and human CD26.

The authors used surface plasmon resonance (SPR) to determine the binding affinities of the MjHKU4r, the HKU4, and the MERS-CoV RBDs to the CD26 of 18 different animal species (including human and other primates, Malayan pangolin, various bat species etc). Site-directed mutagenesis was used to assess the role of individual residues of hCD26, including that of the glycan attached to Asn229.

To determine the host range of the new virus, the authors also made use of homology modeling. The biophysical data revealed that the K_d of the MjHKU4r and MERS-CoV RBDs

to hCD26 is about equal and approx. 8 times lower than that of HKU4. This is in spite of the finding that among the three compared structures, MjHKU4r has the smallest interaction surface with hCD26 (Line 150).

The manuscript offers a very detailed description of the atomic interactions between the three RBDs on the one hand and hCD26 on the other. This is appreciated because it is necessary for understanding the differences in host range of the three viruses although I admit that when reading, I had a tendency to get tired with so much detail... However, Table 2 helps to counteract this effect.

In summary, the work of Zhao et al. demonstrates that the pangolin HKU4-like virus MjHKU4r-CoV-1 binds to human CD26 with relatively high affinity. Together with the broad host range found for this virus, it is conceivable that MjHKU4r-CoV-1 has the potential to be transmitted to humans, especially in view of the fact that pangolins are the most trafficked mammals. Therefore, this manuscript makes an important contribution to getting prepared for a potential MjHKU4r spillover into the human population. It also furthers our knowledge of the details of virus - CD26 interaction.

I have some individual comments:

Lines 25 - 26: Wang et al. (2014) is not the correct reference here. The correct reference is P.C.Y. Woo et al. (2012), which is also in the list of references. This should be corrected. Also, the authors should not state that coronaviruses "could be classified" into alpha, beta, gamma, and delta coronaviruses, as the scientific consensus is to classify them this way. Replace "could be" by "are"...

Response: Thanks for your reminding. They have been corrected.

Lines 26 - 27: This statement is not true for clade A of betacoronaviruses. Only betacoronaviruses of clades B and C can be considered "life-threatening".

Response: Thanks for the comments. It has been rewritten.

Line 27: There have not been "many epidemics and pandemics" caused by CoVs "in the past half-century". A little more accuracy in the description of the history of CoVs would be advisable. The first epidemic caused by a CoV was SARS-CoV in 2003, followed by MERS-CoV in 2012 (and extending over many years), and SARS-CoV-2 (since 2020, and still ongoing). The latter caused the only pandemic among the three. So beta-CoV outbreaks have been recorded since twenty years, not over "half a century".

Response: Thanks for the suggestions. This paragraph has been rewritten.

Line 120: "In the MjHKU4r-CoV-1 RBD/hACE2 complex": Replace "hACE2" by "hCD26".

Response: Thanks for your reminding. It has been corrected.

Line 226: K is the one-letter symbol for lysine, not arginine.

Response: Thanks for your reminding. It has been corrected.

Line 232: "... many VDWs...": This is jargon and should be defined upon first occurrence.

Response: Thanks for your reminding. This jargon was defined in line 122.

The manuscript is written in a reasonable English, although at many places, articles are missing or have been used in the wrong way. There is certainly room for improvement on the language side.

Response: Thanks for the suggestions. This manuscript has been polished by a native speaker.

Dear Prof. Gao,

Thank you for the submission of your revised manuscript to our editorial offices. I have now received the reports from the two referees that I asked to re-evaluate the study, you will find below. As you will see, both referees now supports the publication of the study in EMBO reports. Referee #1 has several suggestions to improve the manuscript and render it more comprehensive, I ask you to address in a final revised manuscript. Please also provide a final p-b-p-response regarding these points.

- Please provide a final title with not more than 100 characters (including spaces).
- Please remove the sentence 'Source data are provided with this paper' from 'Data availability' section, that is restricted to information submitted to external repositories.
- There is a Supplementary Table 1 called out but doesn't exist. Please update or remove the callout.
- Please upload the source data (SD) for the EV figures as one ZIPed folder.

In addition, I would need from you:

Best,

Referee #1 (Referee #2 - TEJ):

The revised manuscript is significantly improved by addition of the binding data for a panel of hCD26 variants. The authors identify which residues on the receptor are important for binding and provide a structure of the RBD-hCD26 complex. My main concern is with the way the data are presented. To appeal to the EMBO reports audience, a more succinct version of results would help, in which a reader would be presented a broader picture of the questions asked and answered found, instead of going into details related to every single residue investigated, which mask the main findings. I propose below some suggestions:

Major revisions:

1. Could the authors summarize their key finding in one sentence in the abstract i.e. spell out what are the crucial host range determinants (like for ex what is said in lines 281-283). If I understand correctly there seem to be 2 residues on hCD26 (aa 295 and 336) that emerge to be the key determinants?
2. Results section, lines 115-132, is heavy for reading and should be simplified and let the associated figure speak for itself. It should also be stated here that N229 mutant of hCD26 could not be produced. The authors state a clear questions, lines 118-119, but after reading the text I was not sure I understood what they found out. A conclusion sentence at the end of this section that answers the posed question would be helpful.
3. Results section, lines 133-203, is too long and heavy with details that this reviewer got lost in. Could the results be presented in a more succinct way? Maybe focus on the number of bonds establishes, and leave it to the reader to check the Table and Figure for details on each residue pair.
4. Lines 156-157 - the authors state an interesting observation that is counter-intuitive to the fact that MjHKU4r RBD binds tighter to hCD26 than the HKU4 and MERS RBDs. Could they explain why that may be at the end of this section. Is it because MjHKU4r RBD forms more bonds / packs better while using fewer residues, or something else. This was not clear to me, and is an important finding I think.
5. Lines 243-245 - the authors present a specific question, but a clear answer (lines 281-283) fails to stand out because this reviewer got lost in the details.

6. Line 297 - sentence "These results indicate that the N-glycans ..." is not appropriate here because the previous sentence states that the N229 hCD26 mutant could not be expressed. So such conclusion is misplaced here.
7. Figure 3A-C: a symbol should be added to illustrate that the first and second row correspond to a certain degree rotation. It is not clear what the 3rd row represents and why the residues are yellow - what is being shown here?
8. Figure 3D and 3E could be used to summarize the findings presented here. For example the residues tested and the key residues in MjHKU4r RBD, identified in this paper, could be highlighted or put in bold to stand out (3E). The same for the residues on the RBD - based on the structural comparisons, were there some that are shared between the 3 analyzed RBDs that are predicted to be very important. Are there some specific for the MjHKU4r RBD that may endow it a higher binding affinity? All of these could be highlighted. It would be helpful for the reader.

Minor revisions:

1. The abstract sentence (lines 14, 15) "It is closely related to bat HKU4-CoV and may achieve direct infection with enhanced infectivity to human cells/organs compared to bat viruses." sounds strange. Do the authors want to say "It is closely related to bat HKU4-CoV and may infect humans more efficiently than bats"?
2. Line 21 "mutagenesis assays" - consider replacing assays with "analyses" or something else
3. Line 231 - this sentence sounds weird, should be rewritten
4. Avoid being vague - line 286 (quite conserved), line 306 (a bit different). Be specific and use numbers.

Referee #2 (Referee #3 - TEJ):

The authors have followed my suggestions for correcting the manuscript and for improving the presentation.

Referee #1:

The revised manuscript is significantly improved by addition of the binding data for a panel of hCD26 variants. The authors identify which residues on the receptor are important for binding and provide a structure of the RBD-hCD26 complex. My main concern is with the way the data are presented. To appeal to the EMBO reports audience, a more succinct version of results would help, in which a reader would be presented a broader picture of the questions asked and answered found, instead of going into details related to every single residue investigated, which mask the main findings. I propose below some suggestions:

Major revisions:

1. Could the authors summarize their key finding in one sentence in the abstract i.e. spell out what are the crucial host range determinants (like for ex what is said in lines 281-283). If I understand correctly there seem to be 2 residues on hCD26 (aa 295 and 336) that emerge to be the key determinants?

Response: Thanks for the suggestions. The abstract has added one sentence summarizing the crucial host range determinants (lines 19-22). Residue sites 291, 292, 294, 295, 336, and 344 of CD26 are the crucial host range determinants for MjHKU4r-CoV-1 (Figs 6 and 7).

2. Results section, lines 115-132, is heavy for reading and should be simplified and let the associated figure speak for itself. It should also be stated here that N229 mutant of hCD26 could not be produced. The authors state a clear questions, lines 118-119, but after reading the

text I was not sure I understood what they found out. A conclusion sentence at the end of this section that answers the posed question would be helpful.

Response: Thanks for the comments. This section has been simplified and stated that the N229 mutants of hCD26 could not be produced (lines 118-119). A conclusion sentence at the end of this section has been included (lines 126-127).

3. Results section, lines 133-203, is too long and heavy with details that this reviewer got lost in. Could the results be presented in a more succinct way? Maybe focus on the number of bonds establishes, and leave it to the reader to check the Table and Figure for details on each residue pair.

Response: Thanks for the suggestions. This section has been simplified.

4. Lines 156-157 - the authors state an interesting observation that is counter-intuitive to the fact that MjHKU4r RBD binds tighter to hCD26 than the HKU4 and MERS RBDs. Could they explain why that may be at the end of this section. Is it because MjHKU4r RBD forms more bonds / packs better while using fewer residues, or something else. This was not clear to me, and is an important finding I think.

Response: Thanks for the suggestions. The AA interaction area of hCD26 in the MjHKU4r-CoV-1 RBD/hCD26 complex is smaller than that of the other two complexes and contains the fewest VDWs. However, the structural alignment shows that MjHKU4r-CoV-1 RBD is closer to hCD26 and its N229 N-glycans than the HKU4-CoV and MERS-CoV RBDs, suggesting that the MjHKU4r-CoV-1 RBD forms stronger interactions to bind more tightly to hCD26 than the other two RBDs (lines 156-161).

5. Lines 243-245 - the authors present a specific question, but a clear answer (lines 281-283)

fails to stand out because this reviewer got lost in the details.

Response: Thanks for the comments. The position of the clear answer has been adjusted ahead of the homology modeling analysis (lines 235-239).

6. Line 297 - sentence “These results indicate that the N-glycans ...” is not appropriate here because the previous sentence states that the N229 hCD26 mutant could not be expressed. So such conclusion is misplaced here.

Response: Thanks for the comments. This sentence has been removed.

7. Figure 3A-C: a symbol should be added to illustrate that the first and second row correspond to a certain degree rotation. It is not clear what the 3rd row represents and why the residues are yellow - what is being shown here?

Response: Thanks for the suggestions. A symbol has been added to illustrate the degree rotation. The figure legend for Figure 3 has been rewritten to describe each panel and label, including 3rd row panels.

8. Figure 3D and 3E could be used to summarize the findings presented here. For example the residues tested and the key residues in MjHKU4r RBD, identified in this paper, could be highlighted or put in bold to stand out (3E). The same for the residues on the RBD - based on the structural comparisons, were there some that are shared between the 3 analyzed RBDs that are predicted to be very important. Are there some specific for the MjHKU4r RBD that may endow it a higher binding affinity? All of these could be highlighted. It would be helpful for the reader.

Response: Thanks for the suggestions. Residues tested in Figure 2 are marked in bold, and key

residues among them are further highlighted with black boxes (Figure 3E). Residues involved in forming H-bonds or hydrophobic interactions with hCD26 and shared by three RBDs are marked in bold. Residues that potentially confer MjHKU4r-CoV-1 RBD with a higher binding affinity are highlighted with black boxes (Figure 3D).

Minor revisions:

1. The abstract sentence (lines 14, 15) “It is closely related to bat HKU4-CoV and may achieve direct infection with enhanced infectivity to human cells/organs compared to bat viruses.” sounds strange. Do the authors want to say “It is closely related to bat HKU4-CoV and may infect humans more efficiently than bats”?

Response: Thanks for the suggestions. It has been rewritten (lines 14-15).

2. Line 21 “mutagenesis assays” - consider replacing assays with “analyses” or something else

Response: Thanks for the suggestions. It has been replaced with “analyses”.

3. Line 231 - this sentence sounds weird, should be rewritten

Response: Thanks for the suggestions. It has been rewritten (lines 223-225).

4. Avoid being vague - line 286 (quite conserved), line 306 (a bit different). Be specific and use numbers.

Response: Thanks for the suggestions. “quite” and “a bit” have been removed.

Prof. George Gao
Institute of Microbiology, CAS
CAS Key Laboratory of Pathogen Microbiology and Immunology
No. 1 Beichen West Road, Chaoyang District
Beijing, Beijing 100101
China

Dear Prof. Gao,

I am very pleased to accept your manuscript for publication in the next available issue of EMBO reports. Thank you for your contribution to our journal.

Yours sincerely,
